# FPTQuant: Function-Preserving Transforms for LLM Quantization

## Abstract

Large language models (LLMs) require substantial compute, and thus energy, at inference time. While quantizing weights and activations is effective at improving efficiency, naive quantization of LLMs can significantly degrade performance due to large magnitude outliers. This paper describes FPTQuant, which introduces three novel, lightweight, and expressive function-preserving transforms (FPTs) to facilitate quantization of transformers: (1) a mergeable pre-RoPE transform for queries and keys, (2) a mergeable transform for values, and (3) a cheap, dynamic per-token scaling transform. By leveraging the equivariances and independencies inherent to canonical transformer operation, we designed these FPTs to maintain the model's function while shaping the intermediate activation distributions to be more quantization friendly. FPTQuant requires no custom kernels and adds virtually no overhead during inference. The FPTs are trained both locally to reduce outliers, and end-to-end such that the outputs of the quantized and full-precision models match. FPTQuant enables static INT4 quantization with minimal overhead and shows SOTA speed-up of up to $3.9\times$ over FP. Empirically, FPTQuant has an excellent accuracy-speed trade-off—it is performing on par or exceeding most prior work and only shows slightly lower accuracy compared to a method that is up to 29% slower.

## 1 Introduction

**Motivation.** Inference on large language models (LLMs) incurs a significant compute toll for every token generated, which ultimately costs money and consumes environmental resources. These costs limit the proliferation of LLM use cases, especially on resource constrained edge devices. They are also a significant barrier to furthering AI research and democratization. Therefore, improving LLM inference efficiency is a critical goal. Of all the numerous LLM efficiency techniques proposed to date, quantization is by far the most successful; significantly reducing the inference cost by reducing the data bit width across the model.

**Transforms for aggressive quantization.** Outliers in transformer weights, activations, and key-value data are a key challenge for quantization (Bondarenko et al., 2021; Kovaleva et al., 2021; Dettmers et al., 2022; Bondarenko et al., 2023; Sun et al., 2024). The fundamental issue is that quantizing outliers to a regular grid leads to an unfortunate range-precision trade-off. We can either (1) capture the outliers by increasing the range, but lose valuable precision at the highest distribution density around zero, or (2) retain precision, but clip the outliers. Both options unfortunately impact model performance. Prior work has explored operations, such as scalings or rotations, that can be added or applied to pretrained networks to smooth outliers without altering the overall model behaviour *in the absence of quantization*. For example, Xiao et al. (2024) take a single linear layer, $\mathbf{W}$, with input, $\mathbf{X}$, and apply a per-channel scaling $\mathbf{T} = \text{diag}(\mathbf{s})$ to $\mathbf{X}$ before quantizing, to reduce outliers, applying the inverse scales to the linear weights. Without quantizers, $(\mathbf{X}\mathbf{T})(\mathbf{T}^{-1}\mathbf{W}) = \mathbf{X}\mathbf{W}$, but with quantizers $Q$, $Q(\mathbf{X}\mathbf{T})Q(\mathbf{T}^{-1}\mathbf{W}) \neq Q(\mathbf{X})Q(\mathbf{W})$. We refer to such operations as *function-preserving transforms (FPTs)*, for which we desire the following properties:

P1 **Function-preservation.** Without any quantization, inserting transform pairs should not change the output (up to computational errors). In practice, this means each FPT typically has an inverse operation.

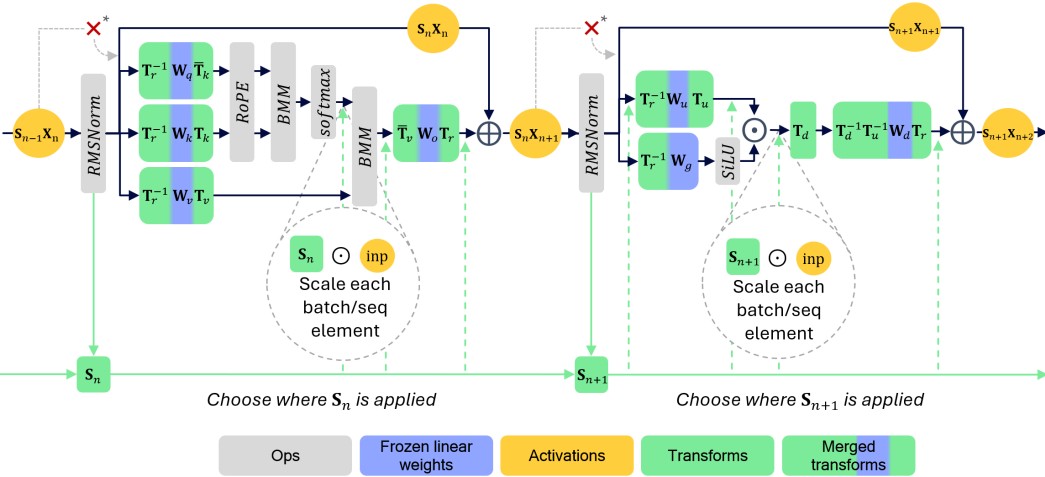

Figure 1: **FPTQuant**. FPTQuant consists of 6 transform types. $(\mathbf{T}_k, \bar{\mathbf{T}}_k)$ is a scale-and-rotate transform merged into the query and key weights; $(\mathbf{T}_v, \bar{\mathbf{T}}_v)$ consists of invertible matrices per head merged into value and output weights; transforms $\{\mathbf{S}_n\}_{n=1}^N$ ($N = 2\times$number of transformer blocks for typical LLMs) are per-token scalers applied to the residual and within the attention and MLP blocks. The scales $\mathbf{S}_n$ are computed by existing RMSNorms, and in practice means now the RMSNorm is also applied to the residual (versus the original network, see $\times^*$). We also use $(\mathbf{T}_u, \mathbf{T}_u^{-1})$ a per-channel scaler merged into up and down projection weights similar to (Hu et al., 2025), partly online Hadamard transform $\mathbf{T}_d$ (Ashkboos et al., 2024a) and mergeable rotation matrix $\mathbf{T}_r$ (Liu et al., 2024a).

P2 **Expressivity.** Transforms with a continuous parametrization and more degrees of freedom are desirable. Continuity means transforms can be optimized directly, e.g. using gradient descent. Extra degrees of freedom offer more flexibility for reducing the quantization error.

P3 **Compute overhead.** Depending on the FPT type and location, it may be possible to merge (or 'fuse') it into an existing operation in a pretrained model. Non-mergeable FPTs represent a new op in the computational graph, and incur additional overhead, as well as requiring software and/or hardware support.

**Contributions.** Our contributions are threefold:

1. We introduce FPTQuant: Function-Preserving Transforms for Quantization (Figure 1). FPTQuant includes three novel FPTs that are designed to be both expressive and cheap.

2. We show FPTQuant enables static INT4 quantization with minimal overhead. This provides a SOTA speed-up of up to $3.9\times$ over FP. FPTQuant requires no kernel-level changes.

3. We show FPTQuant has an excellent accuracy-speed trade-off—it is performing on par or exceeding most prior work and only shows slightly lower accuracy compared to a method that is up to 29% slower.

## 2 RELATED WORK

**Quantization**   Neural network quantization has been demonstrated as an effective technique for reducing the model size and improving computational efficiency (Krishnamoorthi, 2018; Nagel et al., 2021). Quantization methods can generally be categorized into post-training quantization (PTQ) and quantization-aware training (QAT) families. PTQ algorithms take a pretrained high precision network and convert it directly into a fixed-point network without the need for the original training pipeline (Banner et al., 2018; Cai et al., 2020; Choukroun et al., 2019; Hubara et al., 2020; Meller et al., 2019; Zhao et al., 2019; Nagel et al., 2019; 2020; Li et al., 2021). These methods are data-free or only require a small calibration dataset, and are generally fast and easy to use. Quantization-aware training (QAT) methods (Gupta et al., 2015; Jacob et al., 2018; Esser et al., 2020; Bhalgat et al., 2020; Nagel et al., 2022) simulate quantization during training, allowing the model to find more optimal solutions compared to PTQ. However, they generally require longer training times, increased memory usage, need for labeled data and hyperparameter tuning.

**LLM quantization** The excessive training cost and memory usage of traditional QAT methods make them less suitable for quantizing modern LLMs. A few works focus on developing efficient variants of QAT for LLMs include (Liu et al., 2024b; Du et al., 2024; Chen et al., 2024; Dettmers et al., 2024; Xu et al., 2023; Bondarenko et al., 2024). Notably, ParetoQ (Liu et al., 2025) is the only work we are aware of that scales QAT to billions of tokens.

Post-training quantization of LLMs is a challenging task due to presence of strong numerical outliers in weights and activations (Bondarenko et al., 2021; Kovaleva et al., 2021; Dettmers et al., 2022; Bondarenko et al., 2023; Sun et al., 2024). Various strategies have been explored at tackling these difficulties. These include employing second-order information to mitigate the quantization error (Frantar et al., 2022); emphasizing the importance of so-called "salient" weights that correspond to high-magnitude activations (Dettmers et al., 2023; Lin et al., 2023; Lee et al., 2024); separating outliers and use mixed-precision (Kim et al., 2023; Huang et al., 2024; Egiazarian et al., 2024). Some of the other LLM PTQ methods include (Jeon et al., 2023; Lee et al., 2023; Luo et al., 2023; Chee et al., 2024). Note that many of these PTQ techniques focus primarily on weight quantization and memory size reduction.

**Function-preserving transformations** Nagel et al. (2019) explored the idea of FPTs for CNN quantization, observing that ReLU and per-channel scaling commute, which allows scaling of weights across different layers. In the context of LLMs, Xiao et al. (2024) observe that activation quantization is harder than weight quantization due to more outliers. They propose migrating problematic outliers from the activations to the weights, using an online per-channel scaling factor for activations going into linear layers. Wei et al. (2023) add a shift to the scaling, and use a grid search to find a scaler that minimizes the mean-squared error per linear layer. Shao et al. (2024) extend this by including scaling vectors for queries and keys, and using gradient descent to minimize the error per transformer block.

Chee et al. (2024) were the first to consider transforms that mix channels, albeit only for weight quantization, focusing on vector quantization (Tseng et al., 2024) in later work. QuaRot (Ashkboos et al., 2024a) shows randomized Hadamard transforms (RHTs) are effective at reducing outliers. SpinQuant (Liu et al., 2024a) shows that different RHTs perform very differently, yet they cannot be optimized. They extend QuaRot by adding two unconstrained rotation matrices, which are trained to minimize the standard causal LM loss. Critically, these rotation matrices are placed such that they can be merged with weights post-training, negating inference cost. Lin et al. (2024) use online rotations consisting of fixed channel permutations and block diagonal rotations. OSTQuant (Hu et al., 2025) use combinations of scaling vectors and rotations. Recently, FlatQuant (Sun et al., 2025) introduced matrix multiplications with a Kronecker product of two smaller matrices. This provides a transform that is both optimizable, and theoretically cheap to compute. In Appendix A we summarize the associated costs for various transforms and an in-depth comparison of transforms in prior work.

## 3 METHOD

### 3.1 TRANSFORMS

**We argue equivariances and independencies in pretrained models are key to developing better FPTs, and should be explicitly exploited.** Where a candidate FPT is equivariant w.r.t. pretrained model operations, we have the freedom to choose whether to apply it before, or after said operation. This can also influence whether the operation is mergeable. For example, Ashkboos et al. (2024b) used the equivariance $\mathrm{RMSNorm}(\mathbf{XM}) = \mathrm{RMSNorm}(\mathbf{X})\mathbf{M}$ for orthogonal $\mathbf{M}$, to apply a rotation matrix to the residual of LLMs, merging the transform and its inverse into the linear layers of each transformer block. This is a powerful transform, yet it incurs no compute overhead. Understanding equivariances and independencies in networks is thus essential for finding optimal trade-offs between expressivity (P2) and inference cost P3. In this section, we will discuss three equivariances, and how these offer three novel transforms.

### 3.1.1 PRE-RoPE TRANSFORM (MERGEABLE)

Reducing the bit width of KV cache and queries can significantly reduce memory footprint and computational cost of attention, especially with longer context windows. Unfortunately, we cannot naively merge transforms into the query and key projection weights, because modern LLMs use

RoPE positional encodings (Su et al., 2024) (see Appendix C). We introduce a pair of pre-RoPE transforms $(\mathbf{T}_k, \bar{\mathbf{T}}_k)$, where $\mathbf{T}_k$ is applied to keys and $\bar{\mathbf{T}}_k$ can be interpreted as an inverse of $\mathbf{T}_k$, applied to the queries. The transforms consist of scaled $2 \times 2$ rotation matrices, and applying these to the query and key weights Pre-RoPE, the attention output remains unchanged. For simplicity we first assume a single attention head. Denoting $i, j \in \mathbb{N}$ as the token indices and RoPE applied to queries and keys as function $f : \mathbb{R}^d \times \mathbb{N} \to \mathbb{R}^d$ with $f(\mathbf{x}, i) = \mathbf{x}\mathbf{R}_{\Theta,i}^{d_{head}}$ (see details Appendix C), the following holds:

**Theorem 3.1.** *Let $N = d_{head}/2$, and $\mathbf{R}_n \in O(2)$ and $s_n \in \mathbb{R}$, for $n = 1, ..., N$. Define $\mathbf{T}_k = \mathrm{diag}(\mathbf{s}) \, \mathrm{diag}(\{\mathbf{R}_n\}_{n=1}^N)$ and $\bar{T}_k = \mathrm{diag}(\mathbf{s}^{-1}) \, \mathrm{diag}(\{\mathbf{R}_n\}_{n=1}^N)$. Given query and key weights $(\mathbf{W}_q, \mathbf{W}_k) \in \mathbb{R}^{d_{in} \times d_{head}}$, define $\tilde{\mathbf{W}}_q = \mathbf{W}_q\bar{\mathbf{T}}_k$ and $\tilde{\mathbf{W}}_k = \mathbf{W}_k\mathbf{T}_k$. Now it holds:*

$$\langle f(\mathbf{x}_i\tilde{\mathbf{W}}_q, i), f(\mathbf{x}_j\tilde{\mathbf{W}}_k, j) \rangle = \langle f(\mathbf{x}_i\mathbf{W}_q, i), f(\mathbf{x}_j\mathbf{W}_k, j) \rangle$$

See Appendix C for the proof. In practice, for multi-head attention and grouped-query attention, we can choose an independent transform for each key head. Assuming there are $H$ key heads and $mH$ query heads for some $m, H \in \mathbb{N}$ ($m = 1$ for standard multihead attention), this means we have $H$ independent transforms as above. For the more typical grouped query-attention ($m > 1$), each key head is attended to by multiple query heads, hence we need to repeat the corresponding $\mathbf{T}_k$ transform across these heads. Generally, we can thus write:

$$\mathbf{s}^{(h)} \in \mathbb{R}^d, \mathbf{R}_n^{(h)} \in O(2), \quad \forall h, n \tag{1}$$

$$\mathbf{T}_k^{(h)} = \mathrm{diag}(\mathbf{s}^{(h)}) \, \mathrm{diag}(\{\mathbf{R}_n^{(h)}\}_{n=1}^N), \tag{2}$$

$$\mathbf{T}_k = \mathrm{diag}(\{\mathbf{T}_k^{(h)}\}_{h=1}^H) \tag{3}$$

$$\bar{\mathbf{T}}_k = \mathrm{diag}(\underbrace{\bar{\mathbf{T}}_k^{(1)}, ..., \bar{\mathbf{T}}_k^{(1)}}_{m\times}, \bar{\mathbf{T}}_k^{(2)}, ..., \bar{\mathbf{T}}_k^{(H)}), \tag{4}$$

### 3.1.2 MULTIHEAD VALUE TRANSFORM (MERGEABLE)

Note that the attention probabilities $\mathbf{A}$ are of shape $(B, mH, l^1, l^2)$ and the values are of size $(B, mH, l^2, d)$. The batched matmul (BMM) multiplies these per sample, head, and token, and sum this over $l^2$. Note $d$ plays no role in this BMM, consequently we are free to apply any invertible transform to the $d$ axis—in particular, for a single head, it holds that for any invertible matrix $\mathbf{T}$, the attention block output does not change upon merging $\mathbf{T}$ as follows: $(\mathbf{A}(\mathbf{X}\mathbf{W}_v))\mathbf{W}_o = (\mathbf{A}(\mathbf{X}(\mathbf{W}_v\mathbf{T})))(\mathbf{T}^{-1}\mathbf{W}_o)$. Note that the different heads in the values are independent, hence we can apply a different transform to each attention head. Newer models use grouped-query attention, which requires a bit of bookkeeping: we need to repeat the inverses per key head, across the corresponding softmax heads. Assuming there are again $H$ value heads (repeated to $mH$ heads) and $mH$ query heads, we can choose any invertible $\mathbf{T}_v^{(h)} \in \mathbb{R}^{d \times d}$, and set:

$$\mathbf{T}_v = \mathrm{diag}(\{\mathbf{T}_v^{(h)}\}_{h=1}^H), \tag{5}$$

$$\bar{\mathbf{T}}_v = \mathrm{diag}(\underbrace{(\mathbf{T}_v^{(1)})^{-1}, ..., (\mathbf{T}_v^{(1)})^{-1}}_{m\times}, (\mathbf{T}_v^{(2)})^{-1}, ..., (\mathbf{T}_v^{(H)})^{-1}), \tag{6}$$

which are merged into respectively $\mathbf{W}_v$ and $\mathbf{W}_o$ weights.

### 3.1.3 PSEUDODYNAMIC RESIDUAL SCALING

In modern transformer blocks, the residual remains unnormalized—i.e. the LayerNorm or RMSNorm that we apply to the input of the attention and FFN blocks, is not applied to the residual. In practice this means each token of the residual can have a vastly different scale and is difficult to quantize. Even if we do not quantize the residual, this implies that changing the residual representations of tokens $i$ and $j$ may require vastly different scales of the output of the attention and FFN blocks if the norm of $i$'s residual different than that of $j$'s. This may for example explain why the output of the SwiGLU in the FFN (i.e. input of $\mathbf{W}_d$) has serious outliers, see (Bondarenko et al., 2023) and Appendix E, and that subsequent blocks can have similar outlier patterns, see Figure 2(a,b).

Quantization could thus be improved if only the residual was normalized. Fortunately, this can be achieved at virtually no cost, without changing the output of the pretrained network. Moreover, we

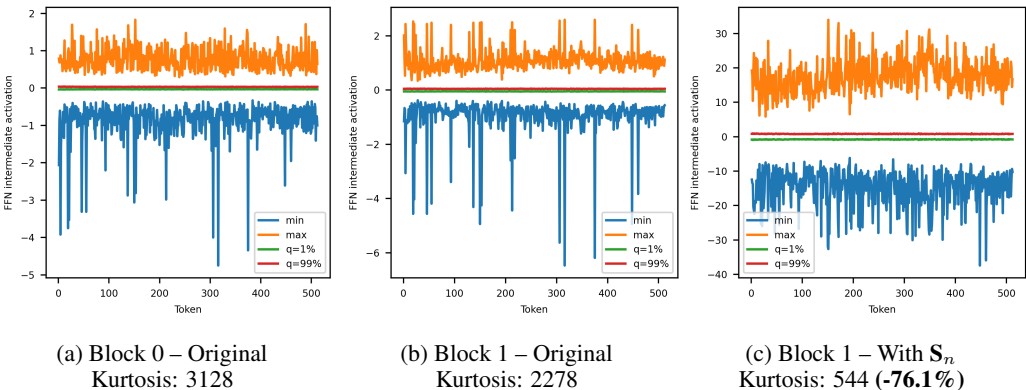

(a) Block 0 – Original
Kurtosis: 3128

(b) Block 1 – Original
Kurtosis: 2278

(c) Block 1 – With $\mathbf{S}_n$
Kurtosis: 544 **(-76.1%)**

Figure 2: **Dynamic scaling reduces intra-block outliers.** We plot the intermediate FFN activations of Llama 3.2 3B-it in the first two blocks (not visualizing the massive [BOS] outlier (Sun et al., 2024)). In the zeroth block, there are some serious tokens with outliers (a). These outliers are absorbed into the residual, and we observe the same tokens cause outliers one block later (b). By applying $\mathbf{S}_1$ (scaling the FFN intermediate layer by the previous residual norm), we see that the outliers are significantly reduced (c).

can apply a similar scaler inside the transformer blocks; this can reduce outliers for intra-block outlier patterns (Figure 2)

**Step 1: move RMSNorm.** Let us index all the blocks in the transformer with $n = 1, ..., N$, where we index the attention and MLP blocks separately (i.e. typically, $N$ equals two times the number of LLM transformer blocks). Let $\mathbf{X}_n$ denote the residual that bypasses a block, $\mathbf{Y}_n$ the output of a block, and $\mathbf{Z}_n = \mathbf{X}_n + \mathbf{Y}_n$. Note, normally $\mathbf{X}_{n+1} = \mathbf{Z}_n$, and the transformer's final output is $\mathbf{Z}_N$. We move the RMSNorm, such that it is applied to the residual too. Let us use $\tilde{\mathbf{X}}_n$ to denote the new residuals. Moving the RMSNorm implies that the residuals are now scaled by a matrix $\mathbf{S}_n = \mathbf{1} \oslash ||\mathbf{X}_n||_R$ of shape (batch, sequence length), where $\oslash$ denotes an element-wise division and $|| \cdot ||_R$ denotes the root-mean-square along the last dimension, $|| \cdot ||_R : \mathbf{x} \mapsto \frac{1}{\sqrt{d}}||\mathbf{x}||_2$. In other words, $\tilde{\mathbf{X}}_n = \mathbf{S}_n \odot \mathbf{X}_n$, with $\odot$ denoting the element-wise multiplication along the dimensions of $\mathbf{S}_n$.

**Step 2: rescale outputs feeding back into residuals.** We do not want to change the network's final output. To ensure this, we need to make sure that anything that feeds back into the residual is rescaled to the new normalized representation. We rescale the outputs $\mathbf{Y}_n$ using the same scales, i.e. ensure:

$$\tilde{\mathbf{Y}}_n = \mathbf{S}_n \odot \mathbf{Y}_n, \tag{7}$$

which then gives $\tilde{\mathbf{Z}}_n = \tilde{\mathbf{X}}_n + \tilde{\mathbf{Y}}_n = \mathbf{S}_n \odot \mathbf{Z}_n$.

Note that (i) matrix multiplication, (ii) linear layers without bias[1], and (iii) BMM all commute with a scaler on the batch/sequence dimension. Consequently, we have a choice where we apply the scale, see Figure 1. Importantly, this means we can apply the rescaling far into the attention and MLP blocks, which we find reduces quantization error within these blocks.

**Computing $\mathbf{S}_n$.** Note that for $n > 1$, $\mathbf{S}_n = \mathbf{1} \oslash ||\mathbf{X}_n||_R = \mathbf{1} \oslash ||\mathbf{Z}_{n-1}||_R = \mathbf{1} \oslash ||\tilde{\mathbf{Z}}_{n-1} \oslash \mathbf{S}_{n-1}||_R = \mathbf{S}_{n-1} \oslash ||\tilde{\mathbf{Z}}_{n-1}||_R$. The right-hand side means we can compute $\mathbf{S}_n$ based on $\tilde{\mathbf{Z}}_{n-1}$, instead of needing to rescale the residual first back to $\mathbf{Z}_{n-1}$ explicitly. We get the recursive relationship:

$$\mathbf{S}_0 = \mathbf{1} \text{ and } \tilde{\mathbf{Z}}_0 = \mathbf{X}_1 \tag{8}$$

$$\mathbf{S}_n = \mathbf{S}_{n-1} \oslash ||\tilde{\mathbf{Z}}_{n-1}||_R \qquad n = 1, ..., N \tag{9}$$

---

[1] We have not found any modern LLMs that use bias for the out and down projection layers.

**Step 3: rescale transformer output (in practice not needed).** ~~Step 3: rescale transformer output (in practice not needed).~~ Note that $\tilde{\mathbf{Z}}_N = \mathbf{S}_n \odot \mathbf{Z}_N$. To ensure we get the same output as the original network, we should divide the very last output by $\mathbf{S}_n$. In practice we do not need to: the transformer is followed by the LM head, which starts with an RMSNorm and hence removes the norm automatically.

### 3.1.4 OTHER TRANSFORMS

In addition to these new transforms, FPTQuant uses a rotation matrix $\mathbf{T}_r$ for rotating the residuals, since this is completely mergeable and effective at reducing activation quantization error (Liu et al., 2024a). Additionally, the notoriously bad activation quantization error at the down projection input (see Appendix E and Table 3 in (Liu et al., 2024a)) warrants an online transform here; we use a Hadamard transform as in (Ashkboos et al., 2024a; Chee et al., 2024), because it is cheap (Table 6). We further use a per-channel scaler transform $\mathbf{T}_u$ that we merge into $\mathbf{W}_u$ and $\mathbf{W}_d$, similar to (Hu et al., 2025), which effectively rescales the channels before the Hadamard transform mixes them. An illustration of all our transforms applied to a typical transformer block is shown in Figure 1.

## 3.2 OPTIMIZATION

### 3.2.1 LOCAL OPTIMIZATION

To reduce the worst outliers, we optimize all transforms first locally and independently—this improves subsequent end-to-end training (Appendix F.2.1). We minimize the $L_p$ norm of each transform's merged weights and use gradient descent. For example, for the residual rotation we optimize:

$$\min_{\mathbf{T}_r} \sum_{i=1}^{\#layers} \Big[ \sum_{\mathbf{W} \in \{\mathbf{W}_q^i, \mathbf{W}_k^i, \mathbf{W}_v^i, \mathbf{W}_u^i, \mathbf{W}_g^i\}} ||\mathbf{T}_r^{-1}\mathbf{W}||_p + \sum_{\mathbf{W} \in \{\mathbf{W}_o^i, \mathbf{W}_d^i, \mathbf{W}_g^i\}} ||\mathbf{W}\mathbf{T}_r||_p \Big], \quad (10)$$

whilst for the PreRoPE transforms $\mathbf{T}_k^i$ of layer $i$, parameterized by $\Phi^i$, we just minimize:

$$\min_{\Phi^i} ||\mathbf{W}_q^i \bar{\mathbf{T}}_k^i||_p + ||\mathbf{W}_k^i \mathbf{T}_k^i||_p.$$

Since $\mathbf{T}_r$ affects all linear layers, we optimize it first (Eq 10). Locally optimized transforms are merged into the weights, after which the next transform is optimized and so forth. We set $p = 4$, following (Bondarenko et al., 2024), who showed $L_4$ is good for determining the quantization grid.

### 3.2.2 END-TO-END OPTIMIZATION

We follow (Liu et al., 2024b) and use student-teacher training for reducing the quantization error further. The original model's weights are frozen. We train the student (the quantized model with transforms) to approximate the teacher (the unquantized FP model), with Jensen-Shannon Divergence loss:

$$\min_{\Phi} \mathbb{E}_X [JSD[f(\mathbf{X}), f_\Phi(\mathbf{X})], \quad (11)$$

where $f$ denotes the original model, $f_\Phi$ the quantized model, and $\Phi$ includes both the transformation and the quantization grid parameters. It is essential we include the latter—the grid cannot adapt to the transformed input otherwise. Note that the original model weights are shared between student and teacher, hence there is no additional memory footprint for the student-teacher framework. In Appendix L we show that FPTQuant's parametrization is stable, i.e. that even with noisy training updates the function-preserving property (P1) holds.

The end-to-end student-teacher approach deviates from SpinQuant (Liu et al., 2024a) and FlatQuant (Sun et al., 2025). SpinQuant uses the LLM's original next-token prediction loss. Compared to next-token prediction, student-teacher training: 1) provides more signal (i.e., for each data point and sequence element, a full vector of probabilities, vs. a single label), and in turn this 2) decreases overfitting. This is an important reason to avoid next-token prediction loss: although we are working with transforms that in the absence of quantization do not change the model output, the combination of the large number of parameters $|\Phi|$ and the quantization non-linearities (i.e. rounding), actually provide the transformed and quantized model with enough capacity to overfit—see Appendix F.2.2. FlatQuant optimizes the mean squared error (MSE) per transformer block. This is not directly applicable for transforms that may affect multiple blocks at once, for example a rotation applied to the residual and merged into all linears, as used here and by (Ashkboos et al., 2024a; Liu et al., 2024a).

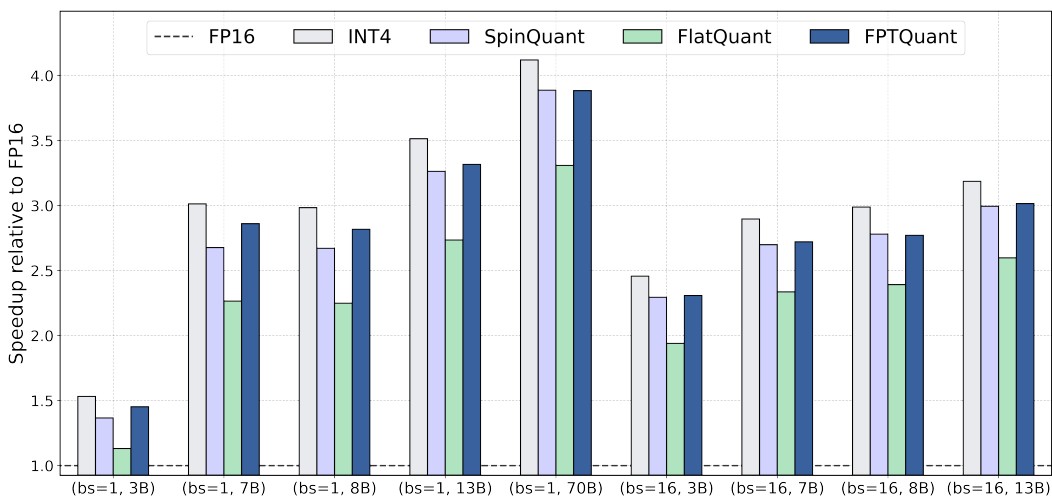

Figure 3: **Static INT4 prefill speedup** of FPTQuant on a single transformer block of Llama across different model sizes (3B, 7B, 8B, 13B, and 70B), batch sizes (1 and 16), with 1024 sequence length.

## 4 EXPERIMENTS

**Evaluation.** We choose a range of models and settings to evaluate FPTQuant. We use Llama 2 7B/13B (Touvron et al., 2023) and Llama 3 8B (Grattafiori et al., 2024) to allow direct comparison to reported results from QuaRot (Ashkboos et al., 2024a), SpinQuant (Liu et al., 2024a) and FlatQuant (Sun et al., 2025). We add to this Llama 3.2 3B instruct—a newer and smaller model that is popular for edge devices. Finally, we test other model families and bigger sizes including Ministral-8B (Mistral.ai, 2025) and Qwen2.5-32B (Yang et al., 2024) . We evaluate on Wikitext-2 (Merity et al., 2017), and use LM-harness to evaluate the same Common Sense Reasoning tasks used in FlatQuant (Sun et al., 2025): PIQA (Bisk et al., 2020), WinoGrande (Sakaguchi et al., 2021), HellaSwag (Zellers et al., 2019), ARC-e and ARC-c (Clark et al., 2018), and LAMBADA (Paperno et al., 2016). In Appendix H we also include results for reasoning tasks (5-shot MMLU and GSM8K).

**Baselines.** We compare FPTQuant against the original floating point model (FP), PTQ using rounding-to-nearest (RTN), RTN with optimizing the quantization ranges (RTN-opt), prior rotation-based including QuaRot (Ashkboos et al., 2024a),SpinQuant (Liu et al., 2024a), OSTQuant (Hu et al., 2025), and the more expensive but state-of-the-art FlatQuant (Sun et al., 2025).

**Training Setup.** For fair comparison, we use the same training and compute budget for all methods— 1024 steps with batch size 16 and sequence length 2048. We train on Wikitext-2 (Merity et al., 2017). We found that end-to-end student-teacher training significantly improves generalization over next-token prediction (see Appendix F.2.2), hence choose to use this for the RTN-opt, SpinQuant and QuaRot baselines too. For each method and quantization setup we select hyper-parameters based on the perplexity on the Wikitext-2 validation set.

**Quantization Setup** Transforms help with low-bit activation quantization, hence we study different activation quantization settings—covering realistic deployment settings (Sec. 4.2), different bit widths (Sec. 4.3), and dynamic activation quantization (Sec. 4.5). Weight quantization methods are tangential to FPTQuant, hence we primarily use round-to-nearest (RTN). In Table 4 we study the combination with the more advanced GPTQ (Frantar et al., 2022)

### 4.1 FPTQUANT IS FAST

**Setup.** We evaluate the runtime performance of our method and compare against other methods using FTPs. We implement FPTQuant, SpinQuant, FlatQuant, and INT4 baseline using PyTorch CUDA/12.1 and using INT4 CUTLASS kernels from QuaRot repository[2]. Note that QuaRot and

---

[2]https://github.com/spcl/QuaRot

Table 1: **FPTQuant excels for harder activation quantization settings.** Exploring different activations quantization settings on Llama 3.2 3B instruct. Left easy, right hard. *Linears+KV* is a popular setting and simplest. *+BMM input* also quantizes the BMM inputs (queries and softmax output). *All except residuals* includes all activations except for the residual. We report Wikitext perplexity—see Appendix G for 0-shot performance and more models.

| #Bits $_{(W-A-KV)}$ | Method | Linears+KV | +BMM input | All except residual |
|---|---|---|---|---|
| 16-16-16 | FP16 | *10.48* | *10.48* | *10.48* |
| | SpinQuant | 11.71 | 10.88 | 11.73 |
| 4-8-8 | FlatQuant | 10.68 | 10.68 | 11.49 |
| | **FPTQuant** | 10.78 | 10.56 | 10.99 |
| | SpinQuant | 12.71 | 13.16 | 20.13 |
| 4-4-4 | FlatQuant | 11.38 | 12.30 | 18.60 |
| | **FPTQuant** | 11.71 | 13.99 | 17.17 |

OSTQuant are about as fast as SpinQuant at inference time. QuaRot is slightly slower, since it has an extra Hadamard transform applied to the head dimension (before $\mathbf{W}_o$); OSTQuant is marginally slower, since it uses online smoothing vectors after the RMSNorm (SpinQuant/FPTQuant merge these into linear layers). For all methods we assume static INT4 quantization. All the measurements are conducted on NVIDIA RTX 3080 Ti. We provide all our experiments on a single transformer block as the whole model does not fit on a single GPU for big enough model size and/or the batch size. We repeat each measurement 1000 times and report the mean speedup relative to FP16 baseline. *More details and additional results with using dynamic quantization are in Appendix I.*

**Results.** Figure 3 shows the prefill speedup of FPTQuant across different batch sizes and model sizes. For most configurations, we get 2.8–3.9× speedup over the FP16 implementation, which is significantly faster than prior reported speedups of QuaRot and FlatQuant. The speedup is consistently increasing with model size and batch size, as the computation becomes the main bottleneck. FPTQuant is on par or faster than SpinQuant and consistently faster than FlatQuant, with a relative speedup of 15–29%. FPTQuant is also faster to train, see Appendix J.1. In all cases FPTQuant is within a 5–6% to the INT4 upper bound.

### 4.2 FPTQUANT EXCELS AT HARD DEPLOYMENT SETTINGS

**Motivation.** There is a large decision space when choosing which activations to quantize. Prior works (Ashkboos et al., 2024a; Liu et al., 2024a; Sun et al., 2025) focus on dynamic quantization of linear inputs and KV cache. This deviates from LLM deployment in practice, which typically (i) has better support for static activation quantization (see Appendix B for details); and (ii) quantizes more intermediate activations for better speed, memory footprint (Tan et al., 2024; Shen et al., 2024). In this experiment, we evaluate SpinQuant, FlatQuant, and FPTQuant for different static activation quantization settings on Llama 3.2 3B instruct. *We observe similar behaviour for Llama 3 8B and Qwen 2.5 7B and zero-shot performance (Appendix G).*

**Results.** We observe (Table 1) that FPTQuant performs comparably to baselines for quantization settings with only linear inputs and KV cache quantize. **It excels at the most challenging setting, in which all activations within the attention and MLP block are quantized.** FPTQuant slightly underperforms baselines when queries and keys are quantized to 4 bit, since the Pre-RoPE transform has less capacity to reduce quantizers here than the non-mergeable transforms of SpinQuant ($R_3$) or FlatQuant ($P_h$). In Appendix F we ablate the value of the different transforms used by FPTQuant.

### 4.3 MAIN RESULTS

**Setup.** In the previous section we saw that FPTQuant outperforms baselines comfortably for the most realistic quantization settings. We extend our evaluation to more models and multiple bit-widths for static quantization, focussing on the setting that FPTQuant did *relatively worst*: only *Linears+KV*. *In Appendix H we include standard deviations for a subset of these results, and include reasoning metrics MMLU and GSM8K.*

Table 2: **Static quantization.** Comparison of the perplexity score on WikiText-2 (Merity et al., 2017) and averaged accuracy on 6 Zero-shot Common Sense Reasoning tasks for Llama2-7B (L2-7B), Llama3.2-3B instruct (L3.2-3B it), Llama3-8B (L3-8B), Ministral-8B instruct (M-8B it), Qwen2.5-32B (Q2.5-32B). *SpinQuant did not yet finish training for Q2.5-32B 4-4-4.*

| #Bits | Method | L2-7B | | L3.2-3B it | | L3-8B | | M-8B it | | Q2.5-32B | |
|---|---|---|---|---|---|---|---|---|---|---|---|
| | | Wiki | 0-shot[6] | Wiki | 0-shot[6] | Wiki | 0-shot[6] | Wiki | 0-shot[6] | Wiki | 0-shot[6] |
| (W-A-KV) | | ($\downarrow$) | Avg.($\uparrow$) | ($\downarrow$) | Avg.($\uparrow$) | ($\downarrow$) | Avg.($\uparrow$) | ($\downarrow$) | Avg.($\uparrow$) | ($\downarrow$) | Avg.($\uparrow$) |
| 16-16-16 | FP16 | 5.47 | 69.79 | 10.48 | 65.63 | 5.75 | 73.33 | 6.45 | 74.37 | 4.67 | 75.29 |
| 4-8-8 | RTN | 73.0 | 47.75 | 40.6 | 47.27 | 77.7 | 45.00 | 5.5e3 | 31.02 | 6.83 | 70.28 |
| | RTN-opt | 7.11 | 56.93 | 11.20 | 61.09 | 7.32 | 67.35 | 10.13 | 51.97 | 5.72 | 75.15 |
| | QuaRot | 6.22 | 63.43 | 10.89 | 63.12 | 7.04 | 67.60 | 6.76 | 73.41 | 5.28 | 76.24 |
| | SpinQuant | 5.97 | 66.01 | 11.03 | 63.28 | 6.54 | 71.60 | 6.86 | 72.48 | 5.28 | 75.51 |
| | OSTQuant | 6.49 | 61.85 | 11.05 | 62.48 | 6.56 | 71.46 | 6.82 | 72.87 | 5.28 | 75.96 |
| | FlatQuant | 6.46 | 62.07 | 10.67 | 65.04 | 6.20 | 72.11 | 6.69 | 73.41 | 5.05 | 76.22 |
| | **FPTQuant** | 5.85 | 65.96 | 10.65 | 64.00 | 6.27 | 72.72 | 6.72 | 73.59 | 5.12 | 77.51 |
| 4-8-4 | RTN | 526 | 38.61 | 128 | 40.40 | 127 | 41.46 | 5.1e3 | 30.29 | 8.08 | 65.79 |
| | RTN-opt | 8.04 | 48.09 | 11.57 | 58.92 | 7.78 | 64.73 | 11.10 | 48.26 | 6.31 | 72.28 |
| | QuaRot | 11.91 | 39.71 | 11.09 | 63.18 | 7.29 | 66.71 | 6.93 | 72.36 | 5.39 | 75.26 |
| | SpinQuant | 6.45 | 59.28 | 11.47 | 59.04 | 7.43 | 65.56 | 7.04 | 71.52 | 5.42 | 76.18 |
| | OSTquant | 7.01 | 55.66 | 11.28 | 61.44 | 7.17 | 68.37 | 6.99 | 71.25 | 5.38 | 74.53 |
| | FlatQuant | 5.91 | 66.04 | 10.88 | 63.69 | 6.51 | 70.83 | 6.83 | 72.67 | 5.31 | 76.37 |
| | **FPTQuant** | 6.05 | 62.68 | 11.12 | 62.42 | 6.78 | 69.46 | 7.04 | 71.10 | 5.37 | 75.76 |
| 4-4-4 | RTN | 2.4e3 | 39.13 | 2.2e3 | 29.17 | 1.6e5 | 37.67 | 1.7e5 | 29.33 | 1.8e6 | 29.83 |
| | RTN-opt | 2.2e3 | 29.54 | 59.06 | 31.16 | 543 | 30.04 | 776 | 29.63 | 2.5e3 | 29.85 |
| | QuaRot | 1218 | 30.21 | 12.81 | 54.38 | 19.72 | 42.76 | 8.34 | 64.68 | 7.51 | 68.01 |
| | SpinQuant | 940 | 30.17 | 12.71 | 54.88 | 11.04 | 54.58 | 8.69 | 60.60 | | |
| | OSTQuant | 519 | 30.75 | 13.41 | 52.43 | 9.66 | 56.69 | 10.00 | 50.88 | 8.39 | 66.84 |
| | FlatQuant | 106 | 29.90 | 11.38 | 61.00 | 9.55 | 61.43 | 8.44 | 63.51 | 7.15 | 70.29 |
| | **FPTQuant** | 603 | 29.76 | 11.71 | 59.46 | 9.74 | 52.96 | 8.49 | 63.69 | 6.98 | 70.60 |

Table 3: **An impact of proposed transforms on Llama 3.2 3B it (W4A4KV4)**. For each setting, we tune the LR and select the best one based on validation Wikitext perplexity. We repeat each experiment 3 times and report mean and standard deviation. We report Wikitext perplexity, average 0-shot CSR, and 5-shot MMLU accuracies.

| Transforms | Wiki ($\downarrow$) | 0-shot[6] ($\uparrow$) | MMLU ($\uparrow$) |
|---|---|---|---|
| - | $60.52^{\pm 1.46}$ | $31.79^{\pm 0.62}$ | $24.96^{\pm 0.34}$ |
| $\{\mathbf{T}_d, \mathbf{T}_r, \mathbf{T}_u\}$ | $12.78^{\pm 0.08}$ | $54.50^{\pm 1.43}$ | $35.46^{\pm 1.18}$ |
| $\{\mathbf{T}_d, \mathbf{T}_r, \mathbf{T}_u\}, \mathbf{S}_n$ | $12.57^{\pm 0.32}$ | $55.38^{\pm 0.69}$ | $36.88^{\pm 0.95}$ |
| $\{\mathbf{T}_d, \mathbf{T}_r, \mathbf{T}_u\}, \mathbf{T}_k$ | $12.45^{\pm 0.25}$ | $55.05^{\pm 1.12}$ | $38.33^{\pm 1.09}$ |
| $\{\mathbf{T}_d, \mathbf{T}_r, \mathbf{T}_u\}, \mathbf{T}_v$ | $11.84^{\pm 0.03}$ | $58.34^{\pm 0.25}$ | $41.54^{\pm 0.88}$ |
| **FPTQuant** (all) | $11.80^{\pm 0.07}$ | $58.87^{\pm 0.74}$ | $44.64^{\pm 0.25}$ |

**Results.** See Table 2. Similar to earlier results, FPTQuant almost always outperforms QuaRot and SpinQuant. OSTQuant generally performs poorly—this is likely due to OSTQuant not being strictly function-preserving, and having stability problems (Appendix L). In most cases FPTQuant shows competitive performance to the significantly slower FlatQuant. However, we do note that for the very challenging setup of W4A4KV4 and Llama 2 7B at W4A8KV4 the gap can sometimes be bigger, especially for zero-shot accuracy. Note that FlatQuant with static quantization can sometimes be unstable in the optimization, e.g. their W4A8KV8 Llama 2 7B results are worse than the more difficult W4A8KV4 ones. We explore this instability further in Appendix L, where we do a sensitivity analysis of the transforms. Also note that for W4A8KV8, we outperform FlatQuant, because the mergeable FPTQuant transforms are better at reducing weight quantization error—at W4A8KV8, this is relatively more important, since activation quantization is easier.

### 4.4 ABLATION OF PROPOSED TRANSFORMS

**Setup.** We explore the value of each transform. Let us take Llama 3.2 3B-it and the same quantization setup as before, *Linears+KV*, at W4A4KV4. We subselect different transforms and repeat the experiment for three seeds. *We include more ablations in Appendix F.1.*

**Results.** We find (Table 3) that each of the three transforms $\mathbf{S}_n, \mathbf{T}_k, \mathbf{T}_v$ helps reduce the perplexity and improves the 0-shot CSR and 5-shot MMLU.

Table 4: **Dynamic quantization.** We run the dynamic quantization experiment (W4A4KV4) from FlatQuant (Table 1 and Table 2, (Sun et al., 2025)), reporting their results for baselines (marked [*]). [§]Using sequence length of 2048. FPTQuant is on par or better than most of the baselines, except FlatQuant, yet FlatQuant is up to 29% slower.

| Method | Weight Quantizer | Llama 2 7B | | Llama 2 13B | | Llama 3 8B | |
|---|---|---|---|---|---|---|---|
| | | Wiki ($\downarrow$) | 0-shot[6] Avg.($\uparrow$) | Wiki ($\downarrow$) | 0-shot[6] Avg.($\uparrow$) | Wiki[§] ($\downarrow$) | 0-shot[6] Avg.($\uparrow$) |
| FP16 | - | 5.47 | 69.79 | 4.88 | 72.55 | 6.14 | 73.33 |
| SmoothQuant[*] | RTN | 83.1 | - | 35.9 | - | 210 | - |
| QuaRot[*] | RTN | 8.56 | 57.73 | 6.10 | 66.25 | 10.60 | 61.34 |
| SpinQuant[*] | RTN | 6.14 | 63.52 | 5.44 | 68.56 | 7.96 | 66.98 |
| OSTQuant | RTN | 6.38 | 65.88 | 5.34 | 69.87 | 7.98 | 68.32 |
| FlatQuant[*] | RTN | 5.79 | 67.96 | 5.12 | 71.42 | 6.98 | 71.23 |
| **FPTQuant** | RTN | 5.97 | 66.06 | 5.37 | 69.81 | 7.67 | 68.41 |
| QuaRot[*] | GPTQ | 6.10 | 65.01 | 5.40 | 68.91 | 8.16 | 65.79 |
| SpinQuant[*] | GPTQ | 5.96 | 66.23 | 5.24 | 70.93 | 7.39 | 68.70 |
| OSTQuant | GPTQ | 5.92 | 66.58 | 5.29 | 70.03 | 7.32 | 68.64 |
| FlatQuant[*] | GPTQ | 5.78 | 67.47 | 5.11 | 71.64 | 6.90 | 71.33 |
| **FPTQuant** | GPTQ | 6.07 | 66.44 | 5.35 | 69.97 | 7.60 | 68.70 |

## 4.5 DYNAMIC QUANTIZATION

**Setup.** We repeat the previous experiment with W4A4KV4 in a dynamic quantization setting. This is identical to the FlatQuant setup, from which we report baseline results. For all methods, we experiment with both round-to-nearest (RTN) and GPTQ for weight quantization. Since OSTQuant only reports results with GPTQ, we used their codebase and provided commands [3] to generate results. For OSTQuant and FPTQuant, we repeat each experiment for three seeds and report the median perplexity and zero-shot accuracy.

**Results.** We observe (Table 4) that FPTQuant is consistently on par or better than all baselines except FlatQuant. However, FPTQuant is up to 29% faster than FlatQuant.

## 5 DISCUSSION

**FPTQuant.** When choosing FPTs, there is a trade-off between expressivity (P2) and cost (P3): more expressive transforms can help reduce quantization error, but incur overhead. By understanding commutation properties of existing operations within the transformer, we have designed most of FPTQuant's transforms to be both expressive, yet mergeable into existing weights. In many settings, the FPTs used by FPTQuant provide a good trade-off between accuracy and speed. For some settings, one may prefer to combine FPTQuant with more expressive, non-mergeable transforms. Choosing which FPTs to choose is largely dependent on the model, quantization setting, and resource constraints. In Appendix K we provide some guidelines for practitioners who want to use FPTs for quantizing their own models.

**Limitations.** We evaluated FPTQuant on LLMs from different generations and with different sizes. While challenges and outlier patterns are often similar across different models (Bondarenko et al., 2023; Kovaleva et al., 2021; Dettmers et al., 2022), it cannot be guaranteed that our insights and gains equally translate to all LLMs.

**Societal impact.** We think FPTQuant has significant positive societal impact. FPTQuant empowers the use of smaller bit widths, which reduces computational, energy, and environmental impact of LLMs. Reduced LLM cost and memory footprint could make LLMs more accessible to economically-disadvantaged populations, and could improve inference on edge devices (e.g. smartphones).

---

[3]https://github.com/BrotherHappy/OSTQuant

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

## A DETAILED TRANSFORMS COMPARISON

In Table 6 we include the representation and theoretical cost of existing transforms. In Table 7 we review existing works, the transforms they use, and their placements.

Table 5 & Figure 4: **Activation quantizers**: aliases and locations.

| Alias | Location |
|-------|----------|
| ao | Attention output |
| ap | Attention probabilities |
| aw | Attention weights |
| d | Down projection output |
| g | Gate projection output |
| gs | SiLU output |
| k | Key projection output |
| ke | Key RoPE-embedded |
| mm | Gate $\odot$ up multiplication |
| na | Norm self-attention |
| nm | Norm MLP/FFN |
| o | Output projection output |
| q | Query projection output |
| qe | Query RoPE-embedded |
| ra | Residual addition self-attention |
| rm | Residual addition MLP/FFN |
| u | Up projection output |
| v | Value projection output |

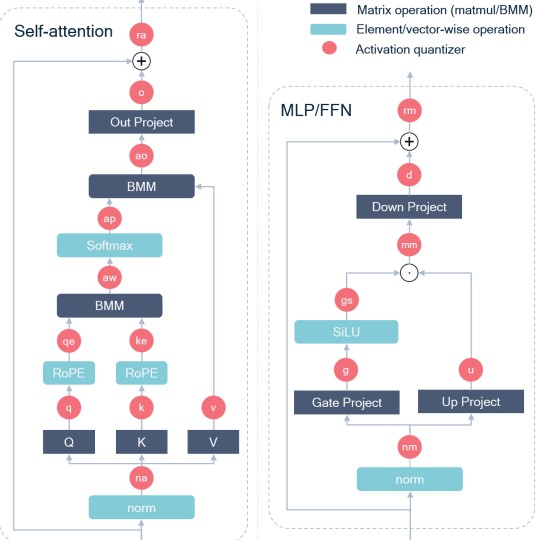

Table 6: **Comparing different transforms.** Cost is measured in terms of a single matrix vector multiplication, $xM$, where $M \in \mathbb{R}^{n \times n}$ and row vector $x \in \mathbb{R}^n$. Memory is total parameters.

| Transform | Cost | Memory | Matrix representation |
|-----------|------|--------|----------------------|
| Scaler | $\mathcal{O}(n)$ | $n$ | $A = \text{diag}(\mathbf{s})$, with $\mathbf{s} \in \mathbb{R}^n$, $s_i \neq 0$ |
| Full matrix | $\mathcal{O}(n^2)$ | $n^2$ | Any invertible matrix $A \in \mathbb{R}^{n \times n}$ |
| Orthogonal | $\mathcal{O}(n^2)$ | $n^2$ | $A \in \mathbb{R}^{n \times n}$ s.t. $AA^T = I$ |
| Rotation | $\mathcal{O}(n^2)$ | $n^2$ | $A \in \mathbb{R}^{n \times n}$ s.t. $AA^T = I$ and $\det(A) = 1$ |
| Block diagonal ($K$ blocks) | $\mathcal{O}(\frac{n^2}{K})$ | $\frac{n^2}{K}$ | $A = \text{diag}(B_1, ..., B_K)$, with invertible $B_k \in \mathbb{R}^{\frac{n}{K} \times \frac{n}{K}}$, $k = 1, ..., K$ |
| Kronecker | $\mathcal{O}(n\sqrt{n})$ | $\sim 2n$ | $P = P_1 \otimes P_2$, with invertible $P_i \in \mathbb{R}^{n_i \times n_i}$ and $n_1 n_2 = n$ (usually $n_1 \approx n_2 \approx \sqrt{n}$) |
| Hadamard Transform (HT) | $\mathcal{O}(n \log n)$ | $0$ | $H_n = \frac{1}{\sqrt{n}} \bigotimes_{i=1}^{\log_2 n} \begin{bmatrix} +1 & +1 \\ +1 & -1 \end{bmatrix}$ |
| Randomized HT (RHT) | $\mathcal{O}(n \log n)$ | $n$ | $\text{diag}(\mathbf{s})H_n$, with Bernoulli $\mathbf{s} \in \{-1, +1\}^n$ |
| Block HT ($K$ blocks) | $\mathcal{O}(n \log[n/K])$ | $0$ | $A = \text{diag}(\{H_{n/K}\}^K)$ |

## B DISCUSSION ON STATIC VS DYNAMIC QUANTIZATION

Traditionally quantization literature and fixed-point accelerators always used static activation scaling factors. However, most LLM quantization literature almost silently started assuming dynamic scaling factors for activations. We call out the distinction here, since in practice it has a big impact on both inference token rate and even which platforms are supported.

**Static quantization** fixes the maximum anticipated range of each quantized tensor ahead of inference time. The quantization grid is determined based on a small calibration dataset, before fixing the scale factors. At runtime, we need only apply these floating-point scale factors following integer matrix multiplication (Nagel et al., 2021). However, despite the ubiquity of static quantization, all prior work to date using FPTs for LLM quantization that we have surveyed (Section 2) assumes dynamic quantization.

Table 7: **Function-preserving transform in LLM quantization literature.** (R)HT: (randomized) hadamard transform. CW: Channel-wise. E2E: end-to-end training, with either original [Label] or student-teacher [ST] loss. For transform locations, see Table 5.

| Work | Transform style | Transform location | Mergeable | Optimization |
|---|---|---|---|---|
| SmoothQuant (Xiao et al., 2024) | CW Scaler | na, nm | True | Local $L_\infty$ |
| Outlier supp+ (Wei et al., 2023) | CW Affine | na, nm | True | Grid search |
| OmniQuant (Shao et al., 2024) | CW Affine
CW Scaler | na, nm, v
(qe, ke) | True
False† | Block-wise
Block-wise |
| QuaRot (Ashkboos et al., 2024a) | HT
HT
RHT | mm‡, ao, (qe, ke)
v
ra, rm | False
True
True | -
-
- |
| SpinQuant (Liu et al., 2024a) | RHT
Rotation | (qe, ke) $R_3$, (mm)
merged into all weights $(R_1)$‡, (v,out) $(R_2)$ | False
True | -
E2E[Label] |
| DuQuant (Lin et al., 2024) | Scaler+Permute+block-wise rotate | linear weights/inputs | False | Iterative greedy |
| OSTQuant* (Hu et al., 2025) | Scaler+orthogonal
HT | na,nm,(v,out)
(qe, ke), mm | True
False | E2E[ST]
- |
| FlatQuant (Sun et al., 2025) | Kronecker

Full
Full | na $(P_v)$, ao $(P_o)$, nm $(P_{ug})$, mm $(P_d)$
(qe, ke) $(P_h)$
(v,out) $(P_v)$ | False

False
True | E2E[Label]

E2E[Label]
E2E[Label] |
| FPTQuant (us)‡ | PreRoPE
Full per head
CW Scaler
Sequence Scaler | (q, k)
(v, out)
(up, down)
(ra, rm, ap, mm) | True
True
True
False | Local $L_p$+E2E[ST]
Local $L_p$+E2E[ST]
Local $L_p$+E2E[ST]
- |

† Authors claim channel-wise scaling of queries and keys can be merged, which does not hold for non-additive positional encodings (e.g. RoPE).
‡ We also use SpinQuant's mergeable $R_1$ rotation, and non-mergeable HT at mm. * OSTQuant also proposed a scaler before RoPE, but this does not commute with RoPE and is thus not function-preserving.

**Dynamic quantization (DQ)** foregoes the calibration step and instead computes scale factors dynamically at runtime for each token independently. This means we can set a large grid for tokens with outliers, while keeping the grid small for tokens without outliers. While this obviously is a huge boon for model performance, it unfortunately introduces a non-trivial compute cost.

**DQ compute overhead.** At inference time, DQ requires the minimum and maximum activation values to be computed and reduced over the last dimension of the whole activation tensor, for each token. The resulting scale factors must then be broadcast and applied to each value. This reduce-broadcast operation can be relatively fast on a CPU, which operates on small chunks of data at a time, such that the binary tree required for the reduction is manageable. However, GPUs and NPUs typically process large tensors at once using custom hardware, and thus the reduction and broadcast tree operations are deep and slow relative to the high throughput MAC operations themselves.

**Lack of support for DQ.** DQ is currently not natively supported on many popular hardware and software stacks. For example, popular quantization packages such as Nvidia TensorRT (Corporation) and PyTorch AO (PyTorch) do not support DQ. Edge hardware platforms also lack support for DQ on their accelerators, including Qualcomm SnapDragon Qualcomm and Nvidia Deep Learning Accelerator (DLA) Nvidia.

## C   PROOF THEOREM 1

**RoPE background.** RoPE's (Su et al., 2024) aim is to modify the queries and keys, such that the output of the query-key multiplication is dependent on their relative positions. RoPE achieves this by multiplying queries and keys with a time-dependent rotation matrix, i.e. RoPE is a function $f : \mathbb{R}^d \times \mathbb{N} \to \mathbb{R}^d$ with $f(\mathbf{x}, i) = \mathbf{x}\mathbf{R}_{\Theta,i}^{d_{head}}$, where $i$ denotes the token index, $\Theta$ the RoPE parameters,

and $d_{head}$ the head dimension. Matrix $\mathbf{R}_{\Theta,i}^{d_{head}}$ is a block-diagonal matrix with $N = d_{head}/2$ blocks. Each block $n$ has size $2 \times 2$ and denotes a rotation of angle $i\theta_n$ of two dimensions. Denoting a 2-dimensional rotation of angle $\theta$ by $\mathbf{R}_\theta^{(2)}$, we can thus write $\mathbf{R}_{\Theta,i}^{d_{head}} = \mathrm{diag}((\mathbf{R}_{i\theta_n})_{n=1}^N)$. As desired, the product between embedded keys and queries depends only on their relative, not absolute, position: $\langle f(\mathbf{q}_i, i), f(\mathbf{k}_j, j) \rangle = \mathbf{q}_i \mathbf{R}_{\Theta,i-j}^d \mathbf{k}_j^\intercal$. We develop transforms that we can apply to queries and keys, yet do not alter the output of the attention softmax. We design these to commute with RoPE's $\mathbf{R}_{\Theta,i}^d$ for all $i$, so that they can be applied before RoPE and merged into $\mathbf{W}_q$ and $\mathbf{W}_k$.

**Theorem 3.1** Let $N = d_{head}/2$, and $\mathbf{R}_n \in O(2)$ and $s_n \in \mathbb{R}$, for $n = 1,...,N$. Define $\mathbf{T}_k = \mathrm{diag}(\mathbf{s}) \, \mathrm{diag}(\{\mathbf{R}_n\}_{n=1}^N)$ and $\bar{T}_k = \mathrm{diag}(\mathbf{s}^{-1}) \, \mathrm{diag}(\{\mathbf{R}_n\}_{n=1}^N)$. Given query and key weights $(\mathbf{W}_q, \mathbf{W}_k) \in \mathbb{R}^{d_{in} \times d_{head}}$, define $\tilde{\mathbf{W}}_q = \mathbf{W}_q \bar{T}_k$ and $\tilde{\mathbf{W}}_k = \mathbf{W}_k \mathbf{T}_k$. Now it holds:

$$\langle f(\mathbf{x}_i \tilde{\mathbf{W}}_q, i), f(\mathbf{x}_j \tilde{\mathbf{W}}_k, j) \rangle = \langle f(\mathbf{x}_i \mathbf{W}_q, i), f(\mathbf{x}_j \mathbf{W}_k, j) \rangle$$

*Proof.* First, let us prove that $\mathbf{T}_k$ commutes with $\mathbf{R}_{\Theta,i}^{d_{head}}$ for any $i$ and $\Theta$. Both are block diagonal (with blocks of size 2×2), so we can treat each block individually. For the individual blocks of $\mathbf{R}_{\Theta,i}^d$ and $\mathbf{T}_k$, write $\mathbf{R}_{i\theta_n}$ and $w_n \mathbf{R}_{\phi_n}$. Trivially, scalars commute with matrices, i.e. $w\mathbf{A} = \mathbf{A}w$ for any matrix $\mathbf{A}$ and $w \in \mathbb{R}$. Additionally, $2 \times 2$ rotations commute, hence $\mathbf{R}_{i\theta_n} w_n R_{\phi_n} = w_n R_{\phi_n} R_{i\theta_n}$. As this holds for all blocks, $\mathbf{R}_{\Theta,i}^{d_{head}} \mathbf{T}_k = \mathbf{T}_k \mathbf{R}_{\Theta,i}^{d_{head}}$.

Second, note that $\bar{\mathbf{T}}_k \mathbf{T}_k^\intercal = I$,[4] since weights and rotations cancel out. Replacing $\mathbf{W}_q, \mathbf{W}_k$ by respectively $\tilde{\mathbf{W}}_q$ and $\tilde{\mathbf{W}}_k$ thus gives attention values:

$$\begin{aligned}
\langle f(\mathbf{x}_i \tilde{\mathbf{W}}_q, i), f(\mathbf{x}_j \tilde{\mathbf{W}}_k, j) \rangle &= \langle \mathbf{x}_i \mathbf{W}_q \bar{\mathbf{T}}_k \mathbf{R}_{\Theta,m}^d, \mathbf{x}_j \mathbf{W}_k \mathbf{T}_k \mathbf{R}_{\Theta,n}^d \rangle \\
&= \langle \mathbf{x}_i \mathbf{W}_q \mathbf{R}_{\Theta,i}^d \bar{\mathbf{T}}_k, \mathbf{x}_j \mathbf{W}_k \mathbf{R}_{\Theta,j}^d \mathbf{T}_k \rangle \\
&= \langle \mathbf{x}_i \mathbf{W}_q \mathbf{R}_{\Theta,i}^d \bar{\mathbf{T}}_k \mathbf{T}_k^\intercal, \mathbf{x}_j \mathbf{W}_k \mathbf{R}_{\Theta,j}^d \rangle \\
&= \langle f(\mathbf{x}_i \mathbf{W}_q, i), f(\mathbf{x}_j \mathbf{W}_k, j) \rangle,
\end{aligned}$$

as desired. $\square$

*Remark* C.1. Note: $\mathbf{R}_{\Theta,i}^d$ overall is a rotation matrix, however rotation matrices generally do not commute unless they share the same axes of rotations. This motivates a transform that uses the same block structure. Note also that a block-wise orthogonal matrix would not suffice, since orthogonal matrices that are not rotations (i.e. that contain also a reflection) do not commute with rotations.

# D   EXPERIMENTAL DETAILS

**General set-up.**   For all experiments and methods, we use batch size 16 with 2048 sequence length for training. We train on Wikitext-2, for 1024 steps with cosine learning rate scheduler, 10% warm-up, and learning rate based on validation PPL. For Qwen 2.5 32B, we use sequence length 512, and 512 steps. In all experiments, we use learnable weight and activation clipping, i.e. the quantization scale and offset are parameters that are updated. We have found significant advantage to optimizing the quantization grid end-to-end, as can be seen by the relatively good RTN-opt baseline in Table 2. For Wikitext perplexity, we evaluate on sequence length 4096—except for Llama 2 7B/13B, for which 2048 is the maximum. We also report assuming sequence length 2048 for Llama 3 8B in Table 4, to allow fair comparison with results from FlatQuant.

**Baselines.**   This work focusses on FPTs. To really understand the value of the FPTs, and not just better and more costly training, we have chosen to use the same training set-up for all methods. This includes optimizing the quantization grid end-to-end. We have found that this significantly improved the performance of some of the baselines—in particular QuaRot and RTN, which do not use any optimization themselves. The only exception is the dynamic quantization experiment (Table 4): to make direct comparison possible, we use the set-up and numbers from FlatQuant, which does not optimize baselines.

---

[4]For single-headed attention, $\bar{\mathbf{T}}_k = \mathbf{T}_k^{-1}$, but this is not true for grouped query attention (Eq. 1 which is typically used in LLMs.

**FPT Parametrization.** We use `torch.nn.utils.parametrizations.orthogonal` to parametrize orthogonal matrices with *Cayley* parametrization. Some FPTs use matrix inversions, e.g. our $\mathbf{T}_v$ and FlatQuant's $P_h$. To avoid computing the inverse during training, it is possible to parametrize these matrices using a singular value decomposition instead, consisting of a diagonal matrix and two orthogonal matrices. In practice, we have found that the added computation of the orthogonality parametrization (which internally computes inverses in any case), leads to slower training and worse results. To avoid potential instability problems with a direct inverse, we choose to keep all transforms in double precision.

**Quantizer range setting.** Although we learn the quantization grids, a good initial quantization grid improves training. We initialize the quantization grid during a range setting stage. For all experiments and method, we pass 64 sequences through the unquantized network and choose a grid that minimizes the $L_p$ norm of the difference between the unquantized and would-be-quantized values. Note that $L_\infty$ corresponds to minmax range setting, which is popular due to its simplicity. In practice, however, we have found that $p = 3$ is better than either minmax, $L_4$ or $L_2$. We choose $L_3$ range setting for all experiments and also for baselines.

**Hadamard non-powers of 2** FPTQuant, SpinQuant(Liu et al., 2024a), and QuaRot (Ashkboos et al., 2024a) use Hadamard transforms. Hadamard transforms are simple to define for powers of 2, namely $H_{2^d} = \bigotimes_{i=1}^{d} \begin{bmatrix} +1 & +1 \\ +1 & -1 \end{bmatrix}$. For some non-powers of 2, there are Hadamard transforms, but these are not implemented in popular packages like `fast-hadamard-transform` (Lab). The simplest approach, and default behaviour in `fast-hadamard-transform`, is to pad with zeros and discard added dimensions after applying the Hadamard transform. This is not correct for FPTs: the added rows are necessary for mapping the transformed activations back to the original values, so setting these rows to zero instead, will yield a different output.

To avoid problems with non-powers of 2, we take a block-wise Hadamard transform: we split dimensions into $K$ groups that are each a power of 2, and apply a standard Hadamard to each. The residual and FFN hidden dimensions $d$ in LLMs are typically $2^n \times K$ with $K$ small—the largest we have found is $K = 43$ for the FFN hidden dimension in Llama 2 7B. The grouped Hadamard can be parallelized efficiently by reshaping the channel dimension into two dimensions of sizes $(K, 2^n)$, applying the Hadamard, and reshaping back. Using a grouped Hadamard reduces the mixing to within groups, but we have found no evidence of a reduced ability to spread outliers due to this—probably because group sizes are still always 256 (for Llama 2's FFN hidden layer) or larger.

# E  QUANTIZATION ERROR PER QUANTIZER

In this section, we study the quantization sensitivity of individual weight and activation quantizers.

**Setup** We apply INT4 RTN quantization, without optimization, to a single quantizer location (see Table 5 for notation) at a time, and report the WikiText-2 test perplexity. We follow the same protocol for the range setting as in the main setup (Appendix D). For activation quantization study, we repeat the experiment three times with different seeds (that affect the random selection of sequences for range estimation) and report the mean value.

**Observations** In Table 8, we can see that each weight quantizer location adds about 0.1 perplexity, on average, while the down projection stands out from the rest a bit more. We can also see that the perplexity drop from quantizing all weights is roughly the sum of drops of individual weight quantizers, meaning that the weight quantization noise is approximately additive.

From Table 9, however, we observe that activation quantization is significantly more challenging, where often a single activation quantizer completely ruins the model performance. Specifically, among the most problematic locations consistently for all models are down projection input/output (mm, d), and residuals (ra, rm). Typically, those locations have the strongest outliers, which makes them difficult to quantize with uniform affine quantization scheme.

Table 8: **Ablation on weight quantizers**. We report WikiText-2 perplexity (lower is better).

| Weight quantizers | Llama 2-7B | Llama 3.2-3B-it | Llama 3-8B | Qwen2.5-7B-it |
|---|---|---|---|---|
| none (FP16) | 5.47 | 10.48 | 5.75 | 6.85 |
| q_proj | 5.567 | 10.434 | 5.795 | 6.914 |
| k_proj | 5.546 | 10.167 | 5.806 | 6.976 |
| v_proj | 5.545 | 10.485 | 5.865 | 6.984 |
| o_proj | 5.504 | 10.628 | 5.859 | 6.952 |
| up_proj | 5.520 | 10.691 | 5.925 | 7.047 |
| down_proj | 5.626 | 11.118 | 6.176 | 7.119 |
| gate_proj | 5.513 | 10.795 | 5.885 | 7.034 |
| all | 6.176 | 11.942 | 6.987 | 7.981 |

Table 9: **Ablation on activation quantizers**. We report WikiText-2 perplexity (lower is better). See Figure 4 for placement of each quantizer.

| Activation quantizers | Llama 2-7B | Llama 3.2-3B-it | Llama 3-8B | Qwen2.5-7B-it |
|---|---|---|---|---|
| none (FP16) | 5.47 | 10.48 | 5.75 | 6.85 |
| ao | 37.9 | 17.1 | 19.6 | 8.10 |
| ap | 1.5e3 | 55.9 | 35.3 | 9.4e3 |
| aw | 6.05 | 12.3 | 6.67 | 4.7e4 |
| d | 8.5e7 | 9.0e3 | 2.3e5 | 1.4e5 |
| g | 36.6 | 25.5 | 29.5 | 9.35 |
| gs | 41.0 | 76.9 | 88.4 | 25.4 |
| k | 5.95 | 12.6 | 6.61 | 3.9e4 |
| ke | 6.02 | 13.6 | 6.90 | 3.3e4 |
| mm | 1.1e4 | 1.7e4 | 3.1e5 | 4.5e4 |
| na | 498 | 101 | 26.3 | 310 |
| nm | 235 | 156 | 122 | 8.2e3 |
| o | 294 | 997 | 1.5e3 | 762 |
| q | 5.71 | 12.3 | 6.83 | 10.9 |
| qe | 5.76 | 12.2 | 6.82 | 12.2 |
| ra | 3.4e4 | 1.3e5 | 1.3e5 | 3.6e4 |
| rm | 3.0e4 | 1.4e5 | 1.3e5 | 8.8e3 |
| u | 34.6 | 31.5 | 43.8 | 13.7 |
| v | 6.70 | 12.0 | 6.65 | 7.13 |
| all | 3.2e4 | 1.3e5 | 1.3e5 | 1.6e5 |

# F ABLATION STUDIES

We introduce FPT $\mathbf{T}_v$ which is an in-place replacement for $R_2$ (SpinQuant) and $P_v$ (FlatQuant). We also propose $\mathbf{T}_k$, which has a similar aim as $R_3$ (SpinQuant) and $P_h$ (FlatQuant), but is mergeable. At last, we introduce $\mathbf{T}_u$, which is an *addition* to QuaRot/SpinQuant's Hadamard transform before the down projection. In this Appendix we ablate these FPTs.

## F.1 TRANSFORM ABLATIONS

$\mathbf{T}_v$. We introduce FPT $\mathbf{T}_v$, which is both mergeable, but also very expressive—we can choose and optimize *any* invertible $d_{head} \times d_{head}$ matrix for each attention head, giving in total $H \times d_{head} \times d_{head}$ degrees of freedom. This is much stronger than SpinQuant's $R_2$ (Liu et al., 2024a), which optimizes a single orthogonal matrix across all value heads (about $d_{head}^2/2$ degrees of freedom). It is also stronger than FlatQuant's $P_v$, who propose two options for parametrizing $P_v$, either a Kronecker or full matrix (see Table 6), but in both cases not chosen per head (max $d_{head} \times d_{head}$ degrees of freedom).

We ablate the value of $\mathbf{T}_v$ compared to $P_v$ (full matrix) and $R_2$. To isolate the effect of these FPTs, we quantize only weights, V-cache, and input to the out projection layer (W4A4). We use the same training set-up as in the main experiments.

We observe (Table 10) that $\mathbf{T}_v$ performs consistently better across models, in particular significantly outperforming SpinQuant's $R_2$. Since all these options have the same inference cost—0, since they are mergeable—we believe $\mathbf{T}_v$ should be a preferred choice.

Table 10: $\mathbf{T}_v$ **is stronger than baseline FPTs.** We compare against $R_2$ and $P_v$ from resp. SpinQuant and FlatQuant, which are also transforms applied to values and mergeable into $\mathbf{W}_v$ and $\mathbf{W}_o$. We use W4A4KV4 with only weights, V-cache, and out projection input quantized, and report Wikitext perplexity (lower is better).

| FPT | L3.2 3B-it | L3 8B | L2 7B |
|---|---|---|---|
| $-$ (RTN-opt) | 11.04 | 7.15 | 5.90 |
| $R_2$ (SpinQuant) | 11.49 | 7.05 | 6.06 |
| $P_v$ (FlatQuant) | 10.86 | 6.67 | 5.74 |
| $\mathbf{T}_v$ (FPTQuant) | 10.82 | 6.63 | 5.73 |

$\mathbf{T}_k$. We conduct a similar ablation for $\mathbf{T}_k$. $\mathbf{T}_k$ is merged into $\mathbf{W}_k$ and $\mathbf{W}_q$, and can thus help with key and query quantization. This is similar to $R_3$ and $P_h$ from respectively SpinQuant and FlatQuant, although these transforms are applied online after the RoPE operator, and thus incur overhead. However, these baselines FPTs are less restricted as a result, and can thus ensure more mixing across channels.

We run a similar experiment as before, only quantizing weights, queries, and keys. We find (Table 11) that for 4-bit quantization of queries and keys, FPTQuant underperforms baselines due to the more restrictive FPT and less mixing across channels. At W4A8, we find $\mathbf{T}_k$ performs on par with baseline FPTs. This experiment clearly shows the expressivity and cost trade-off, P2 vs P3. In some cases, especially when aggressive query-key quantization is beneficial, the overhead of $R_3$ or $P_h$ may weigh up against their higher cost. In Table 12 we show that adding $P_h$ indeed narrows the gap to FlatQuant on the hardest quantization settings.

Table 11: **Ablating Pre-RoPE.** We quantize only weights and post-RoPE queries and keys and compare the performance of three comparable FPTs. We find that the Pre-RoPE transform $\mathbf{T}_k$ underperforms baselines at 4 bit quantization of the queries and keys. This is unsurprising—$\mathbf{T}_k$ is designed to be mergeable before RoPE, but this results in a more constraint, and less expressive FPT. We observe that at 8 bit queries and keys, $\mathbf{T}_k$ performs on par with baselines.

| Quant | FPT | Llama 3.2 3B-it | | Llama 3 8B | | Llama 2 7B | |
|---|---|---|---|---|---|---|---|
| | | Wiki | 0-shot[6] | Wiki | 0-shot[6] | Wiki | 0-shot[6] |
| 4 | $-$ (RTN-opt) | 11.20 | 62.41 | 7.11 | 68.88 | 5.86 | 66.11 |
| | $R_3$ (SpinQuant) | 10.78 | 63.19 | 6.63 | 70.47 | 5.69 | 68.03 |
| | $P_h$ (FlatQuant) | 10.82 | 63.53 | 6.62 | 70.75 | 5.68 | 67.83 |
| | $\mathbf{T}_k$ (FPTQuant) | 11.03 | 62.53 | 6.92 | 69.22 | 5.83 | 66.46 |
| 8 | $-$ (RTN-opt) | 10.71 | 64.59 | 6.45 | 72.06 | 5.64 | 68.56 |
| | $R_3$ (SpinQuant) | 10.70 | 64.42 | 6.44 | 71.04 | 5.64 | 68.27 |
| | $P_h$ (FlatQuant) | 10.71 | 64.88 | 6.44 | 72.00 | 5.65 | 68.26 |
| | $\mathbf{T}_k$ (FPTQuant) | 10.71 | 64.66 | 6.44 | 71.32 | 5.65 | 68.38 |

$\mathbf{T}_u$. The activations before the down projection layer have large outliers. A Hadamard transform at this location has been shown to massively reduce the quantization error (Liu et al., 2024a; Ashkboos et al., 2024a), as it mixes outliers across channels and hence whitens the activation distribution. Whitening is more effective if variables (in this case, channels) have a similar scale. Our scaling transform $\mathbf{T}_u$ achieves exactly this, whilst being completely mergeable.

Table 12: **Including online $P_h$ (Sun et al., 2025) into FPTQuant narrows the gap to FlatQuant.** This incurs some additional overhead, but can be favourable for the hardest quantization settings. We use the same setting as used in Table 2

| #Bits | Method | Llama 3.2 3B-it | | Llama 3 8B | | Llama 2 7B | |
| W-A-KV | | Wiki | 0-shot | Wiki | 0-shot | Wiki | 0-shot |
|---|---|---|---|---|---|---|---|
| 16-16-16 | FP16 | 10.48 | 65.63 | 5.75 | 73.33 | 5.47 | 69.79 |
| 4-8-4 | FlatQuant | 10.88 | 63.69 | 6.51 | 70.83 | 5.91 | 66.04 |
| | FPTQuant | 11.12 | 62.42 | 6.78 | 69.46 | 6.05 | 62.68 |
| | FPTQuant+$P_h$ | 10.81 | 62.91 | 6.63 | 70.12 | 5.98 | 63.04 |
| 4-4-4 | FlatQuant | 11.38 | 61.00 | 9.55 | 61.43 | 951 | 29.70 |
| | FPTQuant | 11.71 | 59.27 | 9.74 | 52.96 | 940 | 29.65 |
| | FPTQuant+$P_h$ | 11.54 | 60.61 | 9.38 | 54.25 | 899 | 29.83 |

In this ablation, we test the performance of a Hadamard transform $\mathbf{T}_d$ with and without $\mathbf{T}_u$. We use a randomized Hadamard transform for $\mathbf{T}_d$, as Liu et al. (2024a) find that even 1 and -1 scales can perform better than non-randomized. Intuitively, $\mathbf{T}_u$ has large benefits over using randomized Hadamard transforms: the randomized Hadamard discrete binary vector is not easy to optimize, does not allow proper scaling down of high-variance channels, and has been shown to exhibit large variance w.r.t. initialization (Liu et al., 2024a). We only quantize the down projection input and weights (W4A4), but leave all other activations unquantized. We train for 512 steps with batch size 8 and sequence length 2048 optimizing quantization grid and $\mathbf{T}_u$ scalers, and run for three seeds.

In Table 13 we observe that adding $\mathbf{T}_u$ has a consistently significant positive effect on quantization error. Like Liu et al. (2024a), we find that the randomized Hadamard transform has large variance, yet we never observe it does better than when $\mathbf{T}_u$ is added. The largest benefit is observed for Llama 2 7B, which has significant outliers in activation before the down projection, which $\mathbf{T}_u$ can scale down.

Table 13: **Adding scaling transform $\mathbf{T}_u$ before the Hadamard transform $\mathbf{T}_d$ significantly reduces quantization error.** Results for W4A4 quantization, with only the input to the down projection quantized. QuaRot and SpinQuant use $\mathbf{T}_d$ only.

| FPT | Llama 3.2 3B-it | | Llama 3 8B | | Llama 2 7B | |
| | Wiki | 0-shot[6] | Wiki | 0-shot[6] | Wiki | 0-shot[6] |
|---|---|---|---|---|---|---|
| $-$ | $121^{\pm 18}$ | $30.63^{\pm 0.35}$ | $4958^{\pm 2399}$ | $29.88^{\pm 0.21}$ | $787^{\pm 160}$ | $29.9^{\pm 0.19}$ |
| $\mathbf{T}_d$ | $12.16^{\pm 0.64}$ | $56.62^{\pm 2.15}$ | $10.75^{\pm 0.62}$ | $60.6^{\pm 0.82}$ | $83.8^{\pm 55.5}$ | $31.15^{\pm 0.84}$ |
| $\mathbf{T}_u, \mathbf{T}_d$ | $10.84^{\pm 0.02}$ | $63.83^{\pm 0.18}$ | $7.5^{\pm 0.23}$ | $67.86^{\pm 0.95}$ | $11.8^{\pm 3.3}$ | $43.13^{\pm 4.95}$ |

## F.2 OPTIMIZATION

### F.2.1 LOCAL OPTIMIZATION

In Section 3.2.1 we proposed a simple data-free and cheap local optimization strategy. Here, we ablate the value of this for overall training stability and speed. We train Llama 3.2 3B-it with FPTQuant end-to-end, with and without first locally optimizing for 200 steps (see Eq. 10). We repeat the experiment for $[0, 32, 128, 256, 512]$ number of end-to-end training steps. As before, we use batch size 16 and sequence length 2048, and 10% warm-up steps.

We observe (Figure 5) that local optimization significantly improves pre-training performance. More importantly, the better initialization advantage persists during end-to-end training, partly due to a more stable training process. With larger number of end-to-end steps, local optimization becomes less beneficial. Locally optimizing all transforms sequentially for 200 steps each takes only about 9 minutes (equivalent in wall time to about 20 end-to-end training steps), and this could be reduced further by parallelizing. Consequently, we find that local optimization is a simple approach to make end-to-end training faster and more efficient.

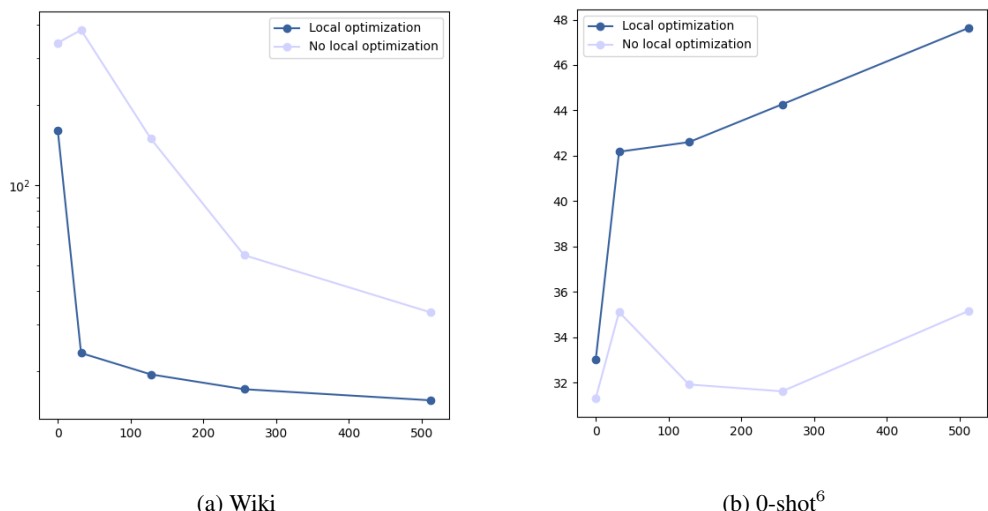

(a) Wiki                              (b) 0-shot[6]

Figure 5: **Local optimization (Section 3.2.1) leads to more stable and faster end-to-end training.** We train FPTQuant on Llama 3.2 3B instruct with and without local optimization, for different number of end-to-end training steps.

**Choosing $p$.** During local optimization, we minimize the $L_p$ of merged weights. In Table 14 we ablate different values of $p$ for the same setting as before, evaluating the performance before and after end-to-end training. We find that no $L_p$ performs significantly better than another—though not using local optimization ("No opt") does significantly worse on average and has a large variance. As before (Figure 5), this shows that local optimization improves training stability, though the choice of $p$ is less important.

Table 14: **Influence of $p$ on local $L_p$ optimization.** We run FPTQuant with different local optimization losses $L_p$ (and without local optimization, *no opt*). We use Llama 3.2 3B-it with the same settings as the main experiment (Section 4.3). We use 3 seeds. *For some runs, only one MMLU evaluation finished. In these cases we leave out the standard deviation.*

| #Bits (W-A-KV) | $p$ | Before end-to-end training | | | After end-to-end training | | |
|---|---|---|---|---|---|---|---|
| | | **Wiki** | **0-Shot[6]** | **MMLU** | **Wiki** | **0-Shot[6]** | **MMLU** |
| 4-8-4 | No opt | $12.70^{\pm0.41}$ | $54.27^{\pm0.45}$ | $28.63$ | $11.18^{\pm0.02}$ | $60.65^{\pm0.10}$ | $47.80$ |
| | 2 | $11.90^{\pm0.03}$ | $56.84^{\pm0.89}$ | $38.74^{\pm1.44}$ | $11.10^{\pm0.02}$ | $61.96^{\pm0.25}$ | $50.21$ |
| | 3 | $11.79^{\pm0.08}$ | $57.75^{\pm0.13}$ | $43.95^{\pm1.20}$ | $11.20^{\pm0.04}$ | $61.43^{\pm0.48}$ | $50.96^{\pm0.62}$ |
| | 4 | $11.83^{\pm0.03}$ | $59.62^{\pm0.54}$ | $45.06^{\pm0.67}$ | $11.19^{\pm0.03}$ | $62.01^{\pm0.34}$ | $50.63^{\pm1.92}$ |
| | 5 | $11.82^{\pm0.03}$ | $58.97^{\pm0.30}$ | $45.94^{\pm1.42}$ | $11.26^{\pm0.05}$ | $61.86^{\pm0.34}$ | $50.45^{\pm1.04}$ |
| | 6 | $11.78^{\pm0.03}$ | $58.23^{\pm0.08}$ | $43.91^{\pm0.46}$ | $11.17^{\pm0.02}$ | $61.83^{\pm0.51}$ | $51.02^{\pm0.25}$ |
| 4-4-4 | No opt | $4114^{\pm321}$ | $29.63^{\pm0.54}$ | $24.55^{\pm0.55}$ | $12.70^{\pm0.41}$ | $54.27^{\pm0.45}$ | $28.63$ |
| | 2 | $340.01^{\pm17.42}$ | $31.36^{\pm0.15}$ | $24.76^{\pm0.24}$ | $11.90^{\pm0.03}$ | $56.84^{\pm0.89}$ | $38.74^{\pm1.44}$ |
| | 3 | $401.93^{\pm35.12}$ | $30.84^{\pm0.15}$ | $24.50^{\pm0.15}$ | $11.79^{\pm0.08}$ | $57.75^{\pm0.13}$ | $43.95^{\pm1.20}$ |
| | 4 | $435.74^{\pm78.71}$ | $30.40^{\pm0.53}$ | $24.84^{\pm0.46}$ | $11.83^{\pm0.03}$ | $59.62^{\pm0.54}$ | $45.06^{\pm0.67}$ |
| | 5 | $340.46^{\pm2.87}$ | $31.10^{\pm0.29}$ | $25.27$ | $11.82^{\pm0.03}$ | $58.97^{\pm0.30}$ | $45.94^{\pm1.42}$ |
| | 6 | $448.20^{\pm41.51}$ | $31.21^{\pm0.10}$ | $24.03$ | $11.78^{\pm0.03}$ | $58.23^{\pm0.08}$ | $43.91^{\pm0.46}$ |

F.2.2  STUDENT-TEACHER TRAINING.

We compare the value of end-to-end training in a student teacher fashion (E2E[ST]), versus the original next-token prediction loss (E2E[label]) used in e.g. SpinQuant (Liu et al., 2024a). We take Llama 3.2 3B instruct and use the same set-up as before—training on Wikitext with sequence length 2048, 1024 training steps, and batch size 16.

We observe (Table 15) that next-token prediction leads to consistently better Wikitext perplexity. This makes sense: the loss for next token prediction is highly similar to the loss of Wikitext perplexity, and since we train and evaluate on (different splits of) Wikitext, the next-token prediction loss fine-tunes the FPT weights and quantization grid to directly minimize this loss. However, we also observe that for FPTs with learnable transforms and hence more capacity (SpinQuant, FPTQuant), **the next-token prediction leads to significantly worse 0-shot performance**, which indicates that the next-token prediction loss generalizes poorly to tasks that are different than the training set. In other words, the next-token prediction loss allows the model to overfit to the target task. Student-teacher training does not allow the same level of overfitting, since the output is fitted to match the whole unquantized output vector. This also has the added value that a whole vector of probabilities (student-teacher training) provides more signal than a one-hot label (next-token prediction).

It may seem counterintuitive that FPTs can lead to such overfitting, since they are designed to preserve the model function (P1). However, note that most FPTs have a large number of trainable parameters (Table 6), which together with a learnable quantization grid, including activation clipping, entails a large capacity to change the function *post-quantization*. For example, next-token prediction could relatively easily decrease the loss by increasing the probability of words that are typical Wikitext lingo (which could be achieved through simple clipping of non-typical tokens). Student-teacher loss would not benefit from this, since even for untypical tokens, it needs to match the output probability.

FPTs are appealing because they do not alter the model's function significantly and do not require significant training. The tendency of next-token prediction to overfit to the training task is undesirable to this end; overfitting alters the model function significantly, and to avoid it we would need to train for longer with more tasks. Consequently, we discourage researchers from using next-token prediction for training FPTs, unless they desire to fine-tune to a specific training set.

Table 15: **Student-teacher training of FPTs is better for generalization than next-token prediction.** We compare two end-to-end training approaches on Llama 3.2 3B instruct (W4A4KV4 static quantization): next-token prediction, e.g., used in SpinQuant, versus student-teacher training. We observe that for learnable FPTs (SpinQuant, FPTQuant), next-token prediction is able to fit the training set (Wikitext) better, leading to lower Wikitext perplexity. However, this does not generalize—the 0-shot common-sense reasoning performance of these models is consistently lower than their student-teacher equivalent.

| Loss | Method | Wiki | 0-Shot[6] |
|---|---|---|---|
| E2E[label] | RTN-opt | **46.29** | **32.98** |
| E2E[ST] | RTN-opt | 46.84 | 31.16 |
| E2E[label] | SpinQuant | **11.23** | 50.58 |
| E2E[ST] | SpinQuant | 12.71 | **54.88** |
| E2E[label] | FPTQuant | **11.58** | 51.73 |
| E2E[ST] | FPTQuant | 11.71 | **59.27** |

# G  QUANTIZATION SETTINGS EXTENDED

We extend Table 1 to include 0-shot performance and extra models. Table 16 includes Llama 3.2 3B instruct and Llama 3 8B, as well as Qwen 2.5 7B instruct (Yang et al., 2024). The latter deteriorates significantly for all transforms at 4-bit activations, due to more challenging activation distributions—see Table 9. Consequently, we use W4A4KV4 for the Llama models, but W4A8KV8 for Qwen.

# H  REASONING PERFORMANCE AND STANDARD DEVIATIONS

To give insight into the stability of methods and significance of results, we repeat the experiment of Table 2 for Llama 3.2 3B instruct for multiple seeds and estimate standard deviation. We also include reasoning metrics 5-shot MMLU and GSM8K. This gives Table 17

Table 16: **FPTQuant does better at harder quantization settings.** Table 1 extended. Exploring different activations quantization settings with W4KV4A4 (Llama 3.2 3B instruct and Llama 3 8B) and W4A8KV8 (Qwen 2.5 7B instruct). *Linears+KV* is the setting used in (Ashkboos et al., 2024a; Liu et al., 2024a; Sun et al., 2025). *+BMM input* also quantizes the inputs to the attention batched matmuls. *All except residual* includes all intermediate activations, except for residual. We see that FPTQuant tends to underform slightly on +BMM, due to the Pre-RoPE transform being cheaper, but less expressive,t than baseline FPTs. This is compensated on the strictest setting, where FPTQuant almost consistently outperforms both baselines.

| Quant | Method | L3.2 3B-it | | L3 8B | | Q2.5 7B-it | |
|---|---|---|---|---|---|---|---|
| | | Wiki | 0-shot[6] | Wiki | 0-shot[6] | Wiki | 0-shot[6] |
| | | ($\downarrow$) | Avg.($\uparrow$) | ($\downarrow$) | Avg.($\uparrow$) | ($\downarrow$) | Avg.($\uparrow$) |
| Linear+KV | Spinquant | 12.73 | 52.85 | 11.04 | 54.58 | 7.66 | 71.95 |
| | FlatQuant | 11.37 | 61.32 | 9.55 | 61.00 | 7.47 | 72.69 |
| | FPTQuant | 12.78 | 54.27 | 9.74 | 59.27 | 7.61 | 71.80 |
| +BMM | Spinquant | 12.47 | 53.96 | 17.57 | 37.84 | 7.87 | 70.74 |
| | FlatQuant | 12.30 | 57.64 | 15.42 | 44.21 | 7.51 | 72.04 |
| | FPTQuant | 13.72 | 49.66 | 12.14 | 45.09 | 7.74 | 69.53 |
| All except residual | Spinquant | 20.83 | 39.94 | 52.27 | 34.04 | 9.23 | 65.95 |
| | FlatQuant | 18.64 | 46.43 | 23.45 | 41.19 | 9.24 | 66.78 |
| | FPTQuant | 16.95 | 44.77 | 18.51 | 41.84 | 8.44 | 68.17 |

Table 17: **More metrics and standard deviations.** Llama 3.2 3B-it W4A4KV4 and W4A8KV4 static quantization, run with 3 seeds to provide an estimate of standard deviation.

| # Bits | Method | Wiki | 0-shot[6] | 5-shot MMLU | GSM8K |
|---|---|---|---|---|---|
| (W-A-KV) | | ($\downarrow$) | ($\uparrow$) | ($\uparrow$) | ($\uparrow$) |
| 16-16-16 | FP16 | 10.48 | 65.63 | 59.69 | 28.20 |
| 4-8-4 | RTN-opt | 11.68±0.07 | 58.65±0.47 | 46.45±1.04 | 12.56±2.13 |
| | QuaRot | 11.03±0.03 | 62.71±0.23 | 51.68±0.50 | 18.52±1.95 |
| | SpinQuant | 11.50±0.07 | 61.96±0.11 | 52.14±0.41 | 22.29±1.02 |
| | FlatQuant | 10.90±0.02 | 63.84±0.70 | 55.46±0.26 | 22.59±0.75 |
| | FPTQuant | 11.06±0.02 | 62.95±0.53 | 52.62±0.48 | 18.57±1.68 |
| 4-4-4 | RTN-opt | 64.86±4.09 | 32.48±0.18 | 24.98±0.18 | 1.14±0.06 |
| | QuaRot | 13.25±0.58 | 51.32±2.41 | 35.72±1.96 | 3.74±1.27 |
| | SpinQuant | 13.04±0.08 | 53.44±0.48 | 38.17±0.73 | 5.61±0.61 |
| | FlatQuant | 11.49±0.06 | 60.07±0.84 | 49.01±1.29 | 17.25±0.34 |
| | FPTQuant | 11.82±0.03 | 59.43±0.45 | 43.71±0.22 | 12.38±0.66 |

Overall, we see that in line with the main paper's results, FPTQuant outperforms baselines SpinQuant and QuaRot almost consistently. Especially on the low bitwidth W4A4KV4, FPTQuant improves over SpinQuant/QuaRot very significantly. The more expensive FlatQuant performs comparably to FPTQuant for Wiki and 0-shot, but outperforms FPTQuant on the more sensitive reasoning tasks.

# I  DETAILED BENCHMARKING RESULTS

In this section, we provide additional details on runtime performance setup and evaluation of our method using dynamic quantization in comparison to other methods using FTPs.

**Setup**  We implement FPTQuant, SpinQuant, FlatQuant, and INT4 baselines (using static and dynamic quantization) using PyTorch CUDA/12.1 and using INT4 CUTLASS kernels from QuaRot repository[5]. All the measurements are conducted on NVIDIA RTX 3080 Ti. We provide all our experiments on a single transformer block as the whole model does not fit on a single GPU for big enough model size and/or the batch size. We repeat each measurement 1000 times and report the mean speedup relative to FP16 baseline.

---
[5] https://github.com/spcl/QuaRot

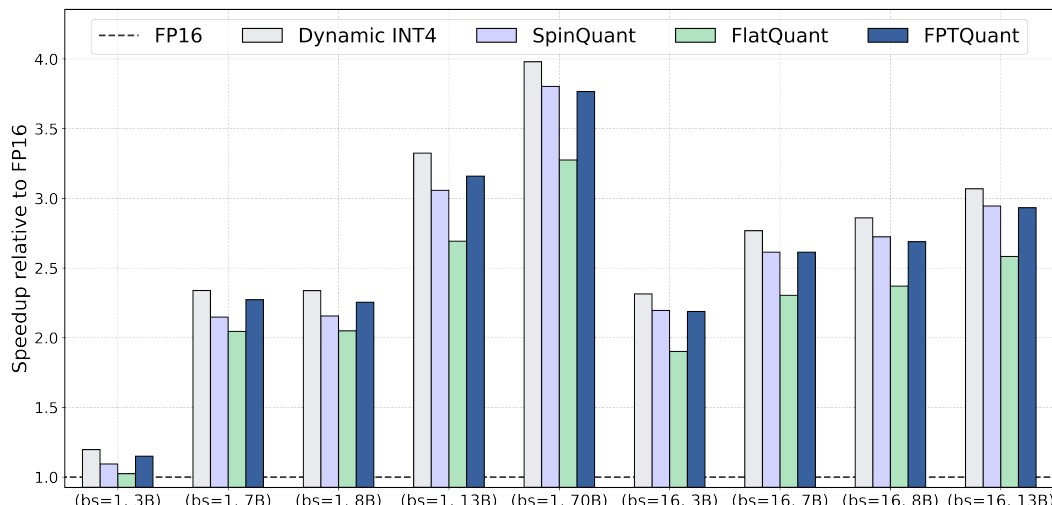

Figure 6: **Dynamic INT4 prefill speedup** of FPTQuant on a single transformer block of LLaMA models across different sizes (3B, 7B, 8B, 13B, and 70B), and batch sizes (1 and 16). We use a sequence length of 1024.

Specifically, we use CUTLASS kernels for quantization/de-quantization and linear layers. Because there is no native INT4 support on Nvidia hardware yet, we use INT8 storage, where each entry represents a pair of INT4 numbers ("double-packed" representation). The kernel for a linear layer, for instance, takes two packed tensors representing weights and activations and computes the matmul assuming INT32 accumulator. Note that query-key and softmax-value BMMs, which are crucial part of the computation, as well as elementwise multiplication in SwiGLU are not quantized in our simulations, and instead are kept in FP16.

**Dynamic INT4 runtime** Figure 6 shows the prefill speedup of FPTQuant across different batch sizes and model sizes, assuming dynamic INT4 quantization. For most configurations, we still get a solid $2.4\times - 3.8\times$ speedup over the FP16 implementation. The speedup is again consistently increasing with model size and batch size, as the computation becomes the main bottleneck. FPTQuant is on par or faster than SpinQuant and consistently faster than FlatQuant, with a relative speedup of 11-21%. Similar to static case, FPTQuant is once again within a 3-6% to the INT4 upper bound.

## J COMPUTE RESOURCES

### J.1 TRAINING COST

Although the FPTQuant transforms are mergeable (except Hadamard transform $\mathbf{T}_d$), we need to consider their training cost.

We detail the training times of the runs from Table 2. For all methods, we trained with a batch size of 4, sequence length 2048, and gradient checkpointing per transformer block, which allowed us to run on a single A100 GPU. We time total training and average over $1024 \times 4$ steps (i.e., 1024 training steps with gradient accumulation of 4), see Table 18.

Table 18: Average training time per step (seconds) across different models and methods.

| Method | Llama 3.2 3B it | Llama 3 8B | Llama 2 7B |
|---|---|---|---|
| RTN-opt | 4.5 | 7.7 | 7.3 |
| QuaRot | 5.3 | 8.7 | 8.4 |
| SpinQuant | 7.9 | 12.0 | 11.6 |
| FlatQuant | 10.3 | 13.7 | 19.5 |
| FPTQuant | 7.1 | 12.6 | 18.3 |

The results are unsurprising. RTN-opt only optimizes the quantization parameters and is trivially the fastest. QuaRot achieves almost the same time—it only adds static Hadamard Transforms to RTN-opt, which incurs virtually no cost.

SpinQuant is significantly more expensive than QuaRot, because it optimizes the relatively high-dimensional rotation matrix $\mathbf{T}_r$ ($R_1$ in their paper). This is more expensive than a standard matrix multiplication at training time, since it requires internally parametrizing the rotation matrix using either Cayley or matrix exponentials (see `torch.nn.utils.parametrizations.orthogonal`). On average, FPTQuant is slightly slower than SpinQuant. This makes sense, considering FPTQuant includes more optimizable transforms in addition to rotation transform $\mathbf{T}_r$. Note that even the most expensive training of FPTQuant (Llama 2 7B for 4096 steps) takes less than 1 single GPU day. Most expensive is FlatQuant, which includes multiple non-mergeable, trainable transforms, including multiple explicit reshapes of activations.

The relative training time of FPTQuant can decrease further with a bit of optimization when sequence length, batch size, or gradient accumulation increase—since all new FPTQuant transforms are mergeable (except the cheap dynamic per-token scaler), they can be merged into the weights prior to passing data and thus do **not** scale with input size. In contrast, FlatQuant transforms almost always have a forward transform applied online at both training and inference time, and thus scale linearly with the input batch size and sequence length.

*Note: local optimization (Section 3.2.1) of FPTQuant is negligible; until convergence takes around 8 minutes for Llama 2 7B (1<% of training).*

### J.2    Training cost of inverse $\mathbf{T}_v$

We may wonder about the cost of the inverse of transform $\mathbf{T}_v$. Surprisingly, the inverse of $\mathbf{T}_v$ is cheap to compute during training. This is partly because of the dimension $d_{\text{head}} = 128$ (for tested models), and partly because it can be computed in parallel across the heads. In Table 19 we time the inverse forward and backward passes, compared to SpinQuant's rotation (parametrized in `torch.nn.utils.parametrizations.orthogonal`). We find that an inverse is cheaper to compute and backpropagate through than parametrized rotations.

**SVD for inverse.**    Large head dimensions can also be supported efficiently by using a singular value decomposition (SVD) to parametrize $\mathbf{T}_v$ during training. For one head $h$, we parametrize:

$$\mathbf{T}_v^h = \mathbf{U} \operatorname{diag}(\mathbf{s}) \mathbf{V}, \quad \text{with } \mathbf{s} \text{ a vector and } \mathbf{U}, \mathbf{V} \in O(d_{\text{head}})$$

This requires rotation reparametrizations and more memory, but the inverse is simpler:

$$\mathbf{T}_v^h = \mathbf{V}^T \operatorname{diag}(\mathbf{s}^{-1}) \mathbf{U}^T$$

In our experiments, however, the SVD parametrization was slower and *not* more accurate than the direct inverse (see Table 19), even for very large head dimensions. This is probably due to the need for two orthogonal matrix parametrizations. Hence, in all the paper's experiments we use the direct inverse.

### J.3    Total compute cost for paper

All the experiments were executed on a single Nvidia A100 GPU equipped with 80GB of VRAM. Models of sizes 3B, 7B and 8B needed respectively around 9.1, 14.5, and 16.5 hours of training with FPTQuant, assuming the main setup with 1024 training steps, sequence length 2048, and a total batch size of $16 = 4$ (per-device batch size) $\times 4$ (gradient accumulation). For obtaining all the results in the paper, including the ablations, we needed 69.8 GPU days (A100). Including preliminary experiments that did not make it in the final paper and hyperparameter tuning we estimate the total compute costs of this research to approximately 386 GPU days.

## K    Guide to choosing FPTs

When choosing FPTs, there is a trade-off between expressivity (P2) and cost ( P3). With FPTQuant, we have aimed to find maximally expressive FPTs that are mergeable or very cheap. FlatQuant is

Table 19: **Benchmarking cost of different transform operations for various sizes.** Benchmarking was performed 1000 times with 5 repeats using `torch.utils.benchmark`, with input size $(1024, 8, \text{dim})$ on an A100. Typical head sizes are 64/128. We observe that the direct inverse is fast. Rotations are slightly slower due to expensive parametrizations (Cayley or matrix exponentials). SVD decomposition for the inverse is even more expensive due to modelling two orthogonal matrices.

| Operation | Dim | Forward (ms) | Forward + Backward (ms) |
|---|---|---|---|
| Inverse (direct) | 64 | 0.463±0.009 | **1.405±0.010** |
| Inverse (via SVD) | 64 | 1.031±0.006 | 2.586±0.011 |
| Rotation (Cayley) | 64 | **0.439±0.003** | 1.523±0.008 |
| Rotation (matrix exp) | 64 | 0.443±0.006 | 2.706±0.030 |
| Inverse (direct) | 128 | 0.623±0.008 | **2.014±0.022** |
| Inverse (via SVD) | 128 | 1.322±0.008 | 3.536±0.020 |
| Rotation (Cayley) | 128 | 0.584±0.002 | 2.200±0.016 |
| Rotation (matrix exp) | 128 | **0.438±0.004** | 2.877±0.013 |
| Inverse (direct) | 1024 | 4.916±0.011 | **34.765±0.329** |
| Inverse (via SVD) | 1024 | 8.611±0.009 | 41.090±0.456 |
| Rotation (Cayley) | 1024 | 4.678±0.004 | 35.869±0.526 |
| Rotation (matrix exp) | 1024 | **1.454±0.005** | 51.086±2.286 |

a strong baseline that regularly outperforms FPTQuant, although this incurs a cost. Fortunately, in practice we can choose on a case-by-case basis which FPTs to include. Here we provide a high-level guide to adding FPTs to your own model.

1. *Explore.* Evaluate quantization error per quantizer placement (e.g. Appendix E)

2. *Choose transforms.* Based on step 1, choose which FPTs to add:

    (a) *Attention and FFN input.* $R_1$ (SpinQuant) and $P_a, P_d$ (FlatQuant) are similar transforms. The first is shared across all layers of the model, whilst FlatQuant's are not. However, an orthogonal matrix $R_1$ has about $d^2/2$ degrees of freedom, whereas each of FlatQuant's Kronecker transforms only has about $2d$ degrees of freedom.[6] Additionally, $R_1$ is mergeable, whereas $P_a$ and $P_d$ are not. As a result of this, $R_1$ should have preference unless a per-layer independent FPT like $P_a, P_d$ is warranted—e.g. if some layers have much higher quantization error than others.

    (b) *Keys and queries.* Depending on how difficult queries and keys are to quantize, one can choose $\mathbf{T}_k$, $R_3$ (SpinQuant), or $P_h$ (FlatQuant), in increasing order of expense and power (see Appendix F.1)

    (c) *Values.* The $\mathbf{T}_v$ transform is more expressive than baselines and as a result better at reducing quantization error (Appendix F). Since it is mergeable and hence free, this should always be used for improving value and out projection input

    (d) *Down projection input.* For many networks, these activations are the trickiest to quantize (Appendix E), which usually warrants an online transform (e.g. Hadamard). If a Hadamard is used, adding the mergeable $\mathbf{T}_u$ improves quantization further (Appendix F.1).

    (e) *Residual* A dynamic residual scaler $S$ can aid quantization if the residual has large outliers in particular tokens. There are multiple possible placements for $S$ (Section 3.1.3), e.g. on the softmax output and after SwiGLU.

3. *Initialize FPTs.* Initialize transforms, e.g. as a Welsh-Hadamard matrix or identity.

4. *Locally optimize FPTs.* Locally optimizing transforms improves performance and reduces training time, whilst incurring very little cost (Appendix F.2.1).

5. *Set quantization range.* Set the initial quantization grid, e.g. using $L_3$ minimization (Appendix D). It is important to only set the grid now, so that initialized FPTs can be taken into account when choosing this grid.

---

[6]Of course, this ignores that more degrees of freedom does not necesarily mean the same space of possible transforms is navigated—e.g. FlatQuant does not have an orthogonality constraint. Nonetheless, we have found $R_1$ to perform comparable as $P_a, P_d$.

Table 20: **FPTQuant is function-preserving, and is stable during optimization.** We add i.i.d. Gaussian noise $N(0, \sigma)$ to all transform parameters, keeping parametrization constraints (e.g. orthogonality) intact. We observe that SpinQuant and FPTQuant remain completely constant—even for larger noise, the output of the model remains the same (as desired by P1)

| $\sigma \rightarrow$ | 0 | 0.1 | 0.3 | 1.0 | 3.0 |
|---|---|---|---|---|---|
| | | | *L3.2-1B-it* | | |
| SpinQuant | $13.16^{\pm 0.00}$ | $13.16^{\pm 0.00}$ | $13.16^{\pm 0.00}$ | $13.16^{\pm 0.00}$ | $13.16^{\pm 0.00}$ |
| OSTQuant | $13.16^{\pm 0.00}$ | $13.20^{\pm 0.02}$ | $19.01^{\pm 4.92}$ | $3.1^{\pm 0.7} \cdot 10^4$ | $4.1^{\pm 1.5} \cdot 10^4$ |
| FlatQuant | $13.16^{\pm 0.00}$ | $13.16^{\pm 0.00}$ | $13.18^{\pm 0.02}$ | $5.5^{\pm 7.5} \cdot 10^5$ | $2.6^{\pm 4.5} \cdot 10^2$ |
| **FPTQuant** | $13.16^{\pm 0.00}$ | $13.16^{\pm 0.00}$ | $13.16^{\pm 0.00}$ | $13.16^{\pm 0.00}$ | $13.16^{\pm 0.00}$ |
| | | | *L3.2-3B-it* | | |
| SpinQuant | $11.05^{\pm 0.00}$ | $11.05^{\pm 0.00}$ | $11.05^{\pm 0.00}$ | $11.05^{\pm 0.00}$ | $11.05^{\pm 0.00}$ |
| OSTQuant | $11.05^{\pm 0.01}$ | $11.06^{\pm 0.05}$ | $14.58^{\pm 1.78}$ | $1.4^{\pm 0.5} \cdot 10^4$ | $1.4^{\pm 0.5} \cdot 10^4$ |
| FlatQuant | $11.05^{\pm 0.00}$ | $11.05^{\pm 0.00}$ | $11.63^{\pm 1.07}$ | $1.5^{\pm 2.9} \cdot 10^5$ | $1.3^{\pm 2.5} \cdot 10^5$ |
| **FPTQuant** | $11.05^{\pm 0.00}$ | $11.05^{\pm 0.00}$ | $11.05^{\pm 0.00}$ | $11.05^{\pm 0.00}$ | $11.05^{\pm 0.00}$ |
| | | | *L3 8B* | | |
| SpinQuant | $6.14^{\pm 0.00}$ | $6.14^{\pm 0.00}$ | $6.14^{\pm 0.00}$ | $6.14^{\pm 0.00}$ | $6.14^{\pm 0.00}$ |
| OSTQuant | $6.14^{\pm 0.00}$ | $6.15^{\pm 0.00}$ | $8.02^{\pm 1.12}$ | $2.7^{\pm 1.5} \cdot 10^4$ | $3.2^{\pm 1.4} \cdot 10^4$ |
| FlatQuant | $6.14^{\pm 0.00}$ | $6.14^{\pm 0.00}$ | $6.35^{\pm 0.14}$ | $8.8^{\pm 11} \cdot 10^5$ | $4.8^{\pm 5.6} \cdot 10^5$ |
| **FPTQuant** | $6.14^{\pm 0.00}$ | $6.14^{\pm 0.00}$ | $6.14^{\pm 0.00}$ | $6.14^{\pm 0.00}$ | $6.14^{\pm 0.00}$ |

6. *Train end-to-end.* Train the FPTs and quantization grid end-to-end, with the unquantized outputs as target.

## L  FUNCTION-PRESERVATION AND SENSITIVITY ANALYSIS TO NOISY TRAINING

The function-preserving property (desideratum P1) of FPTs is useful because it reduces the capacity to change the pretrained model's output, and consequently can avoid overfitting to calibration data. We have conducted a sensitivity analysis to show the function-preserving properties of different transforms *without quantization*. This also gives insight into training stability—if the output of the model is stable w.r.t. the parametrization of transforms, it means noisier gradient updates are less likely to lead to unstable training.

We take the initialized transforms, and simulate noisy training dynamics by perturbing the parameters; we add i.i.d. Gaussian noise with standard deviation $\sigma \in \{0, 0.1, 0.3, 1.0, 3.0\}$ to each transform parameter. Naturally, we ensure the parametrizations remain correct—i.e. that an orthogonal matrix remains orthogonal (using 'torch.nn.utils.parametrization'). We do not add quantizers, since we want to test desideratum P1. We run it for three model sizes, of 1B, 3B, and 8B parameters, and for 5 seeds. See Table 20.

SpinQuant and FPTQuant are very stable—even when we completely randomize the transform parameters, we are ensured that the function-preserving properties are in fact, preserved, and that output is stable. FlatQuant is theoretically function-preserving, but is less stable: their approach consists of 6 transforms per transformer block, which each have 1 or 2 online matrix inversions. The latter can result in floating point precision issues which destroy the function-preservation, roughly observed from $\sigma = 0.3$ (for Llama 3.2 3B-it) and completely destroying the model performance at $\sigma = 1$. We have found this is not an issue during training as long as a small learning rate is chosen (i.e. the noise is small and the optimizer can correct errors in later steps).

OSTQuant is not function-preserving; it uses smoothing transforms that do not cancel each other out. For example, their $S_{qk}$ transform does not commute with RoPE, and hence the query and key transforms do not cancel out (i.e. no function-preservation). We see this in the results. Smoothing vectors are initialized as identities, hence without noise the model works as expected (yields identical output to the original full precision network). When the transform weights are updated even with

relatively small noise ($\sigma = 0.3$), the model is no longer function-preserving and deviates significantly from the original model. This also means that the model has capacity to overfit the training data.

