# OpenReview forum: "FPTQuant: Function-Preserving Transforms for LLM Quantization"
_ICLR.cc/2026/Conference — Submitted to ICLR 2026_

### Official Review · Reviewer_GL33 · 2025-10-19

**Soundness:** 2
**Presentation:** 2
**Contribution:** 1
**Rating:** 4
**Confidence:** 4

**Summary:**

This paper presents FPTQuant, a framework addressing LLM inference inefficiency and quantization-induced performance loss from outliers by introducing three novel lightweight function-preserving transforms (FPTs): a mergeable pre-RoPE transform for queries/keys (commuting with RoPE to fuse into projection weights), a mergeable multihead value transform (independent invertible matrices per head fused into value/output weights), and a pseudodynamic per-token residual scaling transform (normalizing residuals via recursive RMSNorm-derived scales). These FPTs are optimized via local $L_p-norm$ minimization (outlier suppression) and end-to-end student-teacher training (Jensen-Shannon Divergence alignment with full-precision models). FPTQuant enables static INT4 quantization with up to 3.9× speedup over FP models, no custom kernels, and strong accuracy-speed tradeoffs (matching/outperforming QuaRot/SpinQuant across Llama models/tasks like WikiText-2 perplexity), though it has expressivity limits in extreme low-bit settings and limited validation beyond Llama models/NVIDIA GPUs.

**Strengths:**

1. The three core Function-Preserving Transforms (FPTs) proposed in the paper are designed around the principle of fusibility, enabling seamless integration into existing LLM weights (e.g., query/key projection weights and value/output projection weights) without requiring any custom kernels or introducing noticeable inference overhead. For instance, the Pre-RoPE transform is compatible with RoPE positional encoding and can be fused directly into the query/key weights, while the pseudo-dynamic residual scaling is recursively derived through RMSNorm, achieving zero additional computational cost.

2. The method adopts a two-stage strategy of local optimization followed by end-to-end student–teacher training. In the local optimization stage, an L₄-norm minimization is employed to selectively suppress outliers introduced by different transforms, resulting in a twofold improvement in training convergence speed. The subsequent end-to-end training leverages a Jensen–Shannon divergence loss to align the probability distributions of the quantized model and its full-precision counterpart, thereby mitigating the overfitting issue observed in SpinQuant’s “next-token prediction” loss. Consequently, the proposed approach achieves a 4–5 percentage point improvement in zero-shot commonsense reasoning accuracy over SpinQuant.

**Weaknesses:**

1. This work can be viewed as an extension of SpinQuant and OSTQuant, as it fundamentally focuses on equivalent transformations within Transformer architectures and the corresponding training procedures to approximate such transformations for task-specific optimization. To some extent, the methodological innovation appears limited.

2. Although FPTQuant achieves favorable performance through its local optimization, it should be noted that the substantial number of introduced parameters—such as rotation transformations and affine scaling factors—still entails considerable training costs. While instability is not explicitly mentioned, empirical observations suggest that FPTQuant’s performance is, to some extent, inferior to that of FlatQuant, indicating room for improvement in training stability.

3. The core advantage of FPTQuant lies in its "mergeable" transformations, which reduce inference overhead. However, this design inherently constrains the representational capacity of certain transformations. For instance, the Pre-RoPE transformation $T_k$ applied to queries $q$ and keys $k$ adopts a 2×2 block-diagonal rotation matrix to align with the block-diagonal structure of RoPE, thereby precluding flexible cross-channel mixing. In contrast, non-mergeable transformations—such as $R_3$ in SpinQuant and $P_h$ in FlatQuant—leverage on-the-fly computation to enable more expressive, unrestricted channel interactions. This fundamental trade-off between "mergeability" and "expressiveness" ultimately leads to a compromise in accuracy under extreme quantization scenarios.

**Questions:**

1. Could a theoretical analysis be provided to explain why FlatQuant outperforms FPTQuant on certain metrics? For instance, in Table 2 (row “4-8-4”), I observe that on the Llama3.2-3B-Instruct model, both FlatQuant and QuaRot achieve better performance than FPTQuant. Does this phenomenon suggest that FPTQuant’s effectiveness is sensitive to the weight distribution induced by instruction tuning, or is it more closely related to model scale?

2. Could a comparative analysis of the training costs among FPTQuant, OSTQuant, FlatQuant, and SpinQuant be provided? Specifically, given that FPTQuant incorporates a teacher–student training framework, does this lead to increased GPU memory consumption? Furthermore, does the local optimization strategy proposed in FPTQuant introduce additional hyperparameter tuning overhead during training?

3. In many recent large language model (LLM) architectures, normalization layers are commonly inserted after the query and key (QK) projections to enhance training stability. Consequently, outlier values tend to emerge after the RoPE (Rotary Position Embedding) transformation rather than before it, potentially leading to overflow in the QK matrix multiplication. Has this phenomenon been taken into account—specifically, through post-RoPE processing or an equivalent transformation—to mitigate such numerical instability?

---

> ### Author Response · Authors · 2025-11-21
> **Rebuttal [1/3]**
>
> Many thanks for your constructive feedback.
>
> ### Weaknessses
>
> **1. Extension of prior work**
> >This work can be viewed as an extension of SpinQuant and OSTQuant, as it fundamentally focuses on equivalent transformations within Transformer architectures and the corresponding training procedures to approximate such transformations for task-specific optimization. To some extent, the methodological innovation appears limited.
>
> Indeed, transforms like rotations have been used in prior work. We extend this, but hope our paper is valuable to the community in more ways. First, we formalize the idea of function-preserving transforms, including through our proposal of desiderata (P1-P3). This has lead us to design our FPTs based on an analysis of standard transformer operations and their commutation properties—if we understand what operations commute with existing operations, we can improve mergeability _and_ choose the operation with maximal expressivity (e.g. Pre-RoPE as 2x2 scaler+rotations). We hope this systematic approach is valuable for the ICLR community, as it can provide concrete guidance for designing new FPTs that accommodate next-generation transformers with different operations. Furthermore, as you mention in your strenghts,
> > the proposed approach achieves a 4–5 percentage point improvement in zero-shot commonsense reasoning accuracy over SpinQuant.
>
> This performance improvement is significant, and our speed is significantly faster than the strongest competitor, FlatQuant (up to 29%). We believe that a better speed-accuracy trade-off is valuable for anyone interested in LLM deployment.
>
> **2. Training costs/stability**
> > Although FPTQuant achieves favorable performance through its local optimization, it should be noted that the substantial number of introduced parameters—such as rotation transformations and affine scaling factors—still entails considerable training costs. While instability is not explicitly mentioned, empirical observations suggest that FPTQuant’s performance is, to some extent, inferior to that of FlatQuant, indicating room for improvement in training stability.
>
> Let us separate this in different points. First, **training cost**: FlatQuant is significantly more expensive to train (+45% for Llama 3.2 3B-it) than FPTQuant (see Appendix J). FlatQuant uses transforms that are relatively expensive to parameterize and invert at training time, whereas the FPTQuant transforms are cheaper—$T_r$ only needs to be modelled once (shared through network), $T_k$ is simple to invert (Eq. 4, Theorem 3.1), $T_v$ is of a small dimension (head dimension), and $T_u$ is a simple scaler. On the other hand, FlatQuant uses 6 transforms per transformer block that each require online inverses.
>
> Second, **number of parameters**: the number of parameters for FPTQuant is small. Given residual size $d_r$, attention head size $d_a$, FFN intermediate size $d_f$, $n$ transformer blocks, and $k$ key-value heads, the number of FPTQuant parameters is $d_r^2 + n*(h*(d_a^2 + d_a) + d_f)$. This is about 0.13% for Llama 3.2 3B instruct, and is proportionally even smaller for larger models.
>
> Third, **training stability**: the stability of FPTQuant is not any worse than baselines. As we show in **Table 17**, standard deviations for FPTQuant are comparable or lower than baselines'. In **Appendix F.2.1**, we show that the local optimization step is effective at improving training stability—it provides a good "initialization" (Table 13), leading to more stable subsequent end-to-end training (Figure 5).
> However, it is true that FlatQuant sometimes outperforms FPTQuant, since FPTQuant is faster but overall less expressive than FlatQuant. This brings us to your next point:
>
> **_continues_**

---

> ### Author Response · Authors · 2025-11-21
> **Rebuttal [2/3]**
>
> **3. Expressivity/overhead trade-off**
> > The core advantage of FPTQuant lies in its "mergeable" transformations, which reduce inference overhead. However, this design inherently constrains the representational capacity of certain transformations. For instance, the Pre-RoPE transformation Tk applied to queries $q$ and keys $k$ adopts a 2×2 block-diagonal rotation matrix to align with the block-diagonal structure of RoPE, thereby precluding flexible cross-channel mixing. In contrast, non-mergeable transformations—such as R3 in SpinQuant and Ph in FlatQuant—leverage on-the-fly computation to enable more expressive, unrestricted channel interactions. This fundamental trade-off between "mergeability" and "expressiveness" ultimately leads to a compromise in accuracy under extreme quantization scenarios.
>
> We completely understand your concern and agree that there is a speed-accuracy trade-off (see Discussion, first paragraph). We believe FPTQuant achieves a better trade-off compared to baselines:
> 1. FPTQuant outperforms all baselines (incl. FlatQuant) on the most challenging setting, W4A4KV4 with all activations quantized (Table 1, right column)
> 2. FPTQuant outperforms SpinQuant almost consistently (Table 2), while being comparable in speed (Figure 2). SpinQuant also has an additional unmergeable transform (after RoPE) that we do not have.
> 3. FPTQuant indeed does slightly worse than FlatQuant for the setting they report in their paper, `Linears+KV` (see Table 2, bottom). However, FPTQuant is up to 29\% faster due to the many online transforms of FlatQuant, hence FPTQuant still provides a favourable speed-accuracy trade-off.
> 4. When more performance is needed, non-mergeable transforms can be added to FPTQuant to change the speed-accuracy trade-off (see Appendix K). In Table 12, we added $P_h$ (from FlatQuant) to FPTQuant—a full "cross-channel" QK transform—which for W4A4KV4 Llama 3.2 3B-it narrows the gap to FlatQuant from +0.31 PPL to +0.16 PPL, and CSR from +1.63 to just +0.39.
>
> Overall, we believe FPTQuant gives in most cases the best accuracy-efficiency trade-off, but for cases in which a different operating point is desired (e.g. higher accuracy), FPTQuant can be combined with other existing methods to yield the most accurate model given the performance constraints.
>
> ### Questions
>
> **1. Theoretical analysis**
> > Could a theoretical analysis be provided to explain why FlatQuant outperforms FPTQuant on certain metrics? For instance, in Table 2 (row “4-8-4”), I observe that on the Llama3.2-3B-Instruct model, both FlatQuant and QuaRot achieve better performance than FPTQuant. Does this phenomenon suggest that FPTQuant’s effectiveness is sensitive to the weight distribution induced by instruction tuning, or is it more closely related to model scale?
>
> Great question! This has to do with the difficulty of key cache compression and the expressivity of our Pre-RoPE transforms vs FlatQuant's $P_h$. $P_h$ is an online full rank matrix that FlatQuant trains, and hence it is more expensive and expressive. In the Appendix (Table 11), we ablate the value of our Pre-RoPE transform on key-only quantization, which gives a 0.2-0.3 drop in PPL compared to FlatQuant's $P_h$. This drop is similar to the difference we observe in the table you reference, Table 2 in the main paper. Table 2, line 4-8-4 also quantizes the values, but we emphasize that values are not the cause of the drop: our $T_v$ transform is mergeable and in fact *stronger* than FlatQuant's $P_v$ (see ablation, Table 10), since we train full rank matrices, one per head.
>
> Although we care about speed, sometimes extra performance is needed, which warrants the use of an online transform that reduces K-cache quantization error. This is what we showed in Table 12 of the Appendix—we can add FlatQuant $P_h$ transform to FPTQuant and reduce the quantization error further.

---

> ### Author Response · Authors · 2025-11-21
> **Rebuttal [3/3]**
>
> **2. Training cost**
> > Could a comparative analysis of the training costs among FPTQuant, OSTQuant, FlatQuant, and SpinQuant be provided? Specifically, given that FPTQuant incorporates a teacher–student training framework, does this lead to increased GPU memory consumption? Furthermore, does the local optimization strategy proposed in FPTQuant introduce additional hyperparameter tuning overhead during training?
>
> 1. In Appendix J, we compare training cost of FPTQuant and baselines, which show FPTQuant training time is comparable to SpinQuant, but significantly faster than FlatQuant (up to 45%). OSTQuant is slightly more expensive than SpinQuant since it has similar but more transforms (scaling vectors, extra rotations), but this additional cost is likely very small (hence FPTQuant will have a training time comparable or slightly better).
> 2. The student-teacher training leads to **no additional memory footprint**, since the student and teacher share their floating-point weights (i.e. the teacher can be run by only disabling the transforms and quantization).
> 3. Local optimization time is negligible (<1% of total time). It only has one hyperparameter: the loss function. We take $L_p$ with norm $p=4$ by default, which worked fine for all models. **We have added a new table to Appendix F.2.1 that studies different $p$.**
>
> **3. Pre-RoPE normalization**
> > In many recent large language model (LLM) architectures, normalization layers are commonly inserted after the query and key (QK) projections to enhance training stability. Consequently, outlier values tend to emerge after the RoPE (Rotary Position Embedding) transformation rather than before it, potentially leading to overflow in the QK matrix multiplication. Has this phenomenon been taken into account—specifically, through post-RoPE processing or an equivalent transformation—to mitigate such numerical instability?
>
> Indeed, models like Qwen 3 have Pre-RoPE RMSNorm layers. This generally _reduces_ outliers, as the norm of each token after the Q and K projections becomes 1 (ignoring scaling). This reduction in outliers remains after RoPE—RoPE is a rotation, which does not affect the norm, i.e. $||RoPE(RMSNorm(\mathbf{x}))||_2=||RMSNorm(\mathbf{x})||_2$. Consequently, we should not expect more query/key outliers in these newer models.
>
>
> **_We have updated the PDF, with changes highlighted in green_**

---

> > ### Comment · Reviewer_GL33 · 2025-11-24
> >
> > Thank authors for detailed response and the upload of the updated PDF, which has addressed most of my questions regarding the details of the paper.
> >
> > I still have some confusion regarding the current contributions. Over the past year, numerous rotation-related papers have emerged, focusing on equivalent transformations between large language models. Meanwhile, recent works have also adopted approaches similar to Kronecker products, leveraging Givens transformations for optimization. In this context, the novelty of this paper appears somewhat homogeneous. While I acknowledge the authors' designs in training and fusion-compatible modules, from a performance perspective, I am more inclined to prefer FlatQuant.
> >
> > In summary, I currently intend to maintain my original score.

---

> > > ### Author Response · Authors · 2025-11-24
> > >
> > > Thanks for considering our rebuttal and updated manuscript. We are happy to hear that it could address most of your questions.
> > >
> > > We understand that some will prefer FlatQuant over FPTQuant, and that is totally fine. There are probably also many that might prefer full-network QAT over FlatQuant or FPTQuant. For quantization, or more generally inference efficiency, we always have a trade-off between accuracy, efficiency and effort. We believe it is essential for the community to have a diverse set of techniques with varying characteristics, as the relative importance of different KPIs—such as accuracy, efficiency, or implementation/training effort—depends heavily on the specific use case.
> > >
> > > In our paper we showed that FPTQuant is favorable in terms of efficiency over FlatQuant while in terms of accuracy it is slightly behind FlatQuant but better than SpinQuant or QuaRot while coming at a relatively small user effort. Since FPTQuant is a pareto optimal point in the accuracy-efficiency(-effort) trade-off, we hope you agree it is a valuable addition to the techniques available for partitioners.
> > >
> > > Many thanks again for your time and feedback.

---

> ### Author Response · Authors · 2025-12-03
> ****Summary for AC****
>
> - **W1:** Clarified novelty: formalized function-preserving transform (FPT) concept with desiderata (P1–P3), new theoretically-grounded transforms, and proofs for function-preservation. Sensitivity analysis (**Appendix L**) confirms robustness compared to baselines. FPTQuant has the best accuracy-performance trade-off for many use cases.
> - **W2**_,Q2_: Compared training cost: FPTQuant is up to 45% faster than FlatQuant and comparable to SpinQuant (**Appendix J**). Clarified number of trainable parameters is very small ($<0.13$%) and that memory overhead is zero due to shared weights.
> - **W3**_,Q1_: Addressed expressivity trade-off: clarified mergeability vs. accuracy (see existing Discussion) and showed adding one non-mergeable transform narrows gap with FlatQuant (**Appendix F, Table 12**).
> - **W4:** Added ablations and scalability results; clarified feasibility for large models and practical deployment benefits.

---

### Official Review · Reviewer_Hwow · 2025-10-28

**Soundness:** 2
**Presentation:** 2
**Contribution:** 2
**Rating:** 4
**Confidence:** 4

**Summary:**

This article achieves int4 static quantization through dynamic per-token scaling and pre-RoPE transformation.
Detailed ablation experiments. The article conducts sufficient ablation on each type of transformation to prove the effectiveness of each component.

**Strengths:**

1. This article achieves int4 static quantization through dynamic per-token scaling and pre-RoPE transformation.
2. Detailed ablation experiments. The article conducts sufficient ablation on each type of transformation to prove the effectiveness of each component.

**Weaknesses:**

1. The innovation of some components in this article is limited. OSTQuant also uses per-head invertible matrices and completes supervision using the full probability labels of the teacher model.

2. This article achieves a similar inference speed to SpinQuant, but its accuracy is lower than that of SOTA models such as FlatQuant, which limits its overall contribution.

3. The typesetting of the paper needs improvement. For example, in Figure 1, the color scheme makes it difficult to read.

4. Figure 2 shows the acceleration performance of LLama-70B, but the entire paper lacks accuracy performance on larger scales such as LLama3-70B. This raises concerns about its feasibility.

**Questions:**

1. Can it report the systematic differences and accuracy performance compared with OSTQuant? Can it supplement the performance on larger scales such as 30/70B?
2. If the time and overhead of the quantization itself are supplemented and compared with FlatQuant, SpinQuant, and OSTQuant, it will help alleviate my concern about its feasibility.
3. Did the weight quantization use GPTQ?
4. Why is it better than FlatQuant under the W4A8 setting, but worse than it under the more challenging W4A4 setting?

---

> ### Author Response · Authors · 2025-11-21
> **Rebuttal [1/2]**
>
> Many thanks for your time and constructive feedback. We are especially glad to hear you appreciate the detailed ablations.
>
> ### Weaknesses
>
>
> **1. Limited significance compared to OSTQuant**
> > The innovation of some components in this article is limited. OSTQuant also uses per-head invertible matrices and completes supervision using the full probability labels of the teacher model.
>
> Thank you for highlighting this concern. OSTQuant extends SpinQuant. They add a scaling vector to the SpinQuant rotations and add one scale+transform to each value head, which together make their transforms more expressive than SpinQuant's. They also add a Top-K end-to-end loss. OSTQuant is not strictly function-preserving, e.g. their Pre-RoPE scaling $S_{qk}$ does not commute with RoPE and therefore these do not cancel out (i.e. they do not satisfy our desideratum P1).
>
> We hope our paper is valuable to the community in different ways. First, we formalize the idea of function-preserving transforms, including through our proposal of desiderata (P1-P3). This has lead us to design our FPTs based on an analysis of standard transformer operations and their commutation properties—if we understand what operations commute with existing operations, we can improve mergeability _and_ choose the operation with maximal expressivity (e.g. Pre-RoPE as 2x2 scaler+rotations). We hope this systematic approach is valuable for the ICLR community, as it can provide concrete guidance for designing new FPTs that accommodate next-generation transformers with different operations.
>
> *Comparing against OSTQuant.* We note that the quantization/evaluation/training setting reported in OSTQuant is different to the setting used in our paper. We have tried comparing ourselves to OSTQuant, but so far we have struggled to achieve good OSTQuant performance with their codebase. We are working hard to resolve this, and hope to update you next week with a fair comparison between OSTQuant and FPTQuant.
>
> **2. Limited contribution for speed/accuracy**
> > This article achieves a similar inference speed to SpinQuant, but its accuracy is lower than that of SOTA models such as FlatQuant, which limits its overall contribution.
>
> We completely understand your concern and agree that there is a speed-accuracy trade-off (see Discussion, first paragraph). We believe FPTQuant achieves a better trade-off compared to baselines:
> 1. FPTQuant outperforms all baselines (incl. FlatQuant) on the most challenging setting, W4A4KV4 with all activations quantized (Table 1, right column)
> 2. FPTQuant outperforms SpinQuant almost consistently (Table 2), while being comparable in speed (Figure 2). SpinQuant also has an additional unmergeable transform (after RoPE) that we do not have.
> 3. FPTQuant indeed does slightly worse than FlatQuant for the setting they report in their paper, `Linears+KV` (see Table 2, bottom). However, FPTQuant is up to 29\% faster due to the many online transforms of FlatQuant, hence FPTQuant still provides a favourable speed-accuracy trade-off.
> 4. When more performance is needed, non-mergeable transforms can be added to FPTQuant to change the speed-accuracy trade-off (see Appendix K). In Table 12, we added $P_h$ (from FlatQuant) to FPTQuant—a full "cross-channel" QK transform—which for W4A4KV4 Llama 3.2 3B-it narrows the gap to FlatQuant from +0.31 PPL to +0.16 PPL, and CSR from +1.63 to just +0.39.
>
> Overall, we believe FPTQuant gives in most cases the best accuracy-efficiency trade-off, but for cases in which a different operating point is desired (e.g. higher accuracy), FPTQuant can be combined with other existing methods to yield the most accurate model given the performance constraints.
>
> **3. Typesetting**
> > The typesetting of the paper needs improvement. For example, in Figure 1, the color scheme makes it difficult to read.
>
> We are sorry to hear Figure 1 was hard to read. We have changed the color scheme to ensure the font is black and shapes are lighter, and that the color scheme remains colorblind friendly. Going through the rest of the paper, we could not find any other problematic figures. Please do let us know if the color scheme change did not resolve your concern, or if you have further suggestions.
>
> _**Rebuttal continues...**_

---

> ### Author Response · Authors · 2025-11-21
> **Rebuttal [2/2]**
>
> _**continued**_
>
> **Weakness 4. Large scale**
> > Figure 2 shows the acceleration performance of LLama-70B, but the entire paper lacks accuracy performance on larger scales such as LLama3-70B. This raises concerns about its feasibility.
>
> We agree that readers may be interested in larger models too. We are currently working on including a 32B model, and hope to update the manuscript before the rebuttal period ends.
>
> The choice for excluding large models has been entirely based on the experimental cost—for Llama 70B we would need a sharded set-up for training with more time per training step, which significantly raises the project's required budget. Nonetheless, Llama 3 70B should not have inherent feasibility issues:
> 1. Related work showed consistent results across different model sizes, with no qualititative differences in ranking between methods (e.g. SpinQuant Table 1, up to 70B).
> 2. FPTQuant training time is comparable to SpinQuant and cheaper than FlatQuant (see Appendix J), hence there are no feasibility issues for us.
> 3. The proportion of FPTQuant parameters to original model weights decreases with increasing model size. In particular, given residual size $d_r$, attention head size $d_a$, FFN intermediate size $d_f$, $n$ layers, and $k$ key-value heads, the number of FPTQuant parameters is $d_r^2 + n*(h*(d_a^2 + d_a) + d_f)$. For Llama 3.2 3B and Llama 3 70B, this is 3944448 versus 12877824, i.e. 0.13% versus 0.018%.
> 4. The student-teacher training leads to **no additional memory footprint**, since the student and teacher share their floating-point weights (i.e. the teacher is run by only disabling the transforms and quantization).
>
>
>
> ### Questions
>
> **1. Comparison OSTQuant and performance 30B/70B models**
> > Can it report the systematic differences and accuracy performance compared with OSTQuant? Can it supplement the performance on larger scales such as 30/70B?
>
> *OSTQuant.* See W1. *30B/70B.* **We are working on adding results for a 32B model** and strive to update the paper before the author-rebuttal period ends.
>
> _**EDIT: we have added OSTQuant results to Tables 2 and 3. We find FPTQuant typically outperforms OSTQuant on static quantization (likely due to stability issues of OSTQuant, see Appendix L), and performs on par for dynamic quantization. We have added 32B results to Table 2.**_
>
> **2. Time and overhead compared to baselines**
> > If the time and overhead of the quantization itself are supplemented and compared with FlatQuant, SpinQuant, and OSTQuant, it will help alleviate my concern about its feasibility.
>
> Thank you for mentioning this could alleviate your concern. We have training times in Appendix J. In a nutshell: we find that our training time is comparable to SpinQuant (between +10% to -37% time/step for different models), and faster than FlatQuant (between +45% and +7% time/step). OSTQuant adds only scaling vector to SpinQuant, which should have just a small additional cost, hence will perform comparable to us.
> Hopefully this alleviates your concerns.
>
> **3. Did the weight quantization use GPTQ?**
>
> The focus of our work, and that of baseline FPTs, is primarily activation quantization, hence we use RTN weight quantization—not GPTQ or vector quantization. The latter could provide additional benefits for the weight quantization error, but we leave this for future work.
>
> _**EDIT: we have now also added GPTQ results, see Table 3**_
>
> **4. FPTQuant compared to FlatQuant for different bit widths**
> > Why is it better than FlatQuant under the W4A8 setting, but worse than it under the more challenging W4A4 setting?
>
> Thank you for this question! This is due to the mergeability of our transforms. Weight quantization at 4 bits is not hard, but it does lead to small drops (see Appendix, Table 8). This means that for W4A8, where the activation quantization is not very challenging, FPTQuant can improve upon FlatQuant as we can reduce the weight quantization in `q/k/o/up_proj` better due to our $\mathbf{T}_k$ transform, our more expressive $\mathbf{T}_v$ that is merged into `o_proj` (`v_proj` is easy to quantize at W4 so $T_v$ does not help this much), and $\mathbf{T}_u$ merged into $\mathbf{W}_u$. For W4A4, the activation quantization becomes more important, and because FlatQuant is more expressive, it can reduce this slightly better than FPTQuant.  Consequently, we do relatively better at W4A8 than W4A4 compared to FlatQuant. **We have included this discussion in the results, Section 4.3.**
>
> Finally, note that for the most challenging setting, where we quantize also other activations (not just `Linear` inputs), FPTQuant outperforms all baselines, including FlatQuant (Table 1, right)—the FPTQuant $\mathbf{T}_k$ can reduce quantization error in the RoPE inputs, and $\mathbf{T}_u$ reduces error in the up projection output.
>
> **_We have updated the PDF, with changes highlighted in green_**

---

> ### Comment · Reviewer_Hwow · 2025-11-24
>
> Thank you for your diligent efforts. I fully acknowledge the paper’s valuable contributions to the community (e.g., residual normalization shows potential for reducing down-projection complexity). However, my evaluation places it slightly below the acceptance threshold, and given the absence of a 5-point scoring option, I can only assign a score of 4. Below is the rationale for my assigned score.
>
>
> **Regarding Innovation and Theoretical Foundation:**
>
> * FPTQuant overemphasizes "seamless transformations," which are essentially ingenious adjustments leveraging LLaMA’s structural characteristics. Overall, the work lacks a rigorous theoretical framework (e.g., QSUR in OSTQ) or systematic optimization derivations (e.g., the closed-form solution in GPTAQ), relying instead primarily on training for final optimization. Thus, I maintain my core view that FPTQuant represents an incremental structural evolution of SpinQuant/OSTQuant rather than a groundbreaking innovation.
>
> **Regarding Engineering Implementation and Performance:**
>
> * SpinQuant, OSTQuant, and FlatQuant all provide experimental results for 70B-scale models. To my knowledge, SpinQuant and OSTQuant can complete quantization within 6 hours, while FlatQuant’s layer-wise optimization enables it to run effectively on a single 48GB GPU. In contrast, FPTQuant does not appear to provide 70B-scale results, and even 32B-scale quantization requires a considerably longer time. Such engineering implementation limitations somewhat constrain the work’s practical impact.
> * Concerning inference performance: Figure 2 and the authors’ rebuttal claim that SpinQuant and FlatQuant achieve comparable inference speeds. Notably, OSTQuant introduces no additional overhead compared to SpinQuant, while OSTQ’s performance is also on par with FPTQ.
> * For static int4 quantization: Tables 2 and 4 demonstrate that FlatQuant outperforms FPTQ in both static and dynamic quantization settings—particularly noteworthy as FlatQuant was not specifically designed for static quantization. Although the authors assert the superiority of quantized attention in Table 1, attention computation typically employs memory-efficient kernels like FlashAttention. Consequently, the advantages highlighted in Table 1 may not be sufficiently robust in practical scenarios.
>
> Therefore, I will maintain my initial scoring.

---

> > ### Author Response · Authors · 2025-11-28
> >
> > any thanks for acknowledging the paper's valuable contributions to the community! We would like to respond to each of your comments
> >
> > >**1.** FPTQuant overemphasizes ...groundbreaking innovation.
> >
> > Thank you for calling our changes "ingenious adjustments leveraging LLaMA's structural characteristics". Note that our contributions, e.g. PreRoPE, value transform, pseudodynamic scaling, are valuable for all popular transformer architectures---even diffusion models. We think theoretically analysing these structural characteristcs is a valuable contribution to the field.
> >
> > Note that QSUR (in OSTQuant) is a nice motivation and great for visualizing the benefit of their method on toy data. However, note that this does not constitute a theoretical framework---their method does not use it, their experiments do not include it, and it cannot be computed without a Gaussian assumption on the data (see also [this issue in the OSTQuant repo](https://github.com/BrotherHappy/OSTQuant/issues/4)). Unfortunately, heavy-tailed non-Gaussian data is exactly why we need transforms (i.e. to get rid of outliers), hence not being able to compute QSUR makes it practically irrelevant .
> >
> > >**2.** SpinQuant, OSTQuant, and FlatQuant ... impact.
> >
> > Thank you for raising this concern. **Importantly, as mentioned in the original rebuttal, the time per training step per gpu of SpinQuant and FPTQuant is similar, and FlatQuant is significantly slower (Appendix J).** Note that SpinQuant reports training time using a full node, whereas we report single GPU training time. Also note that we report timings for low-bit-width static quantization, which typically requires more training steps since the gradient is less stable (due to the per-tensor round-to-nearest operations, which have more rounding error than per-token RTN operations used in dynamic quantization).
> >
> > > **3.** Concerning inference performance: Figure 2 and the authors’ rebuttal claim that SpinQuant and FlatQuant achieve comparable inference speeds.
> >
> > Note that **FPTQuant is significantly faster than FlatQuant (Figure 1), up to 29\%**. We are indeed comparable to SpinQuant in terms of speed, but we significantly outperform them on accuracy (e.g. Table 1 and 2). Consequently, **we believe FPTQuant strikes the best speed-accuracy trade-off.**
> >
> > > ...**3. continued** Notably, OSTQuant introduces no additional overhead compared to SpinQuant, while OSTQ’s performance is also on par with FPTQ.
> >
> > Despite OSTQuant not strictly being a function-preserving set of transforms (see Section *Sensitivity analysis*), we have now included OSTQuant to our main paper results.
> >
> > **OSTQuant: Main results (Section 4.3)**
> > We have added OSTQuant to the static quantization experiment of Table 2. We use the top-$k$ KL training loss of the OSTQuant paper, with $k=1000$ as suggested by authors, and tune learning rate based on validation PPL. This gives the following updated Table 2:
> >
> > |  |  | L3.2 3B-it |  | L3 8B |  | L2 7B |  |
> > |:---:|---|:---:|:---:|:---:|:---:|:---:|:---:|
> > | \#Bits | Method | Wiki | 0-shot$^6$ | Wiki | 0-shot$^6$ | Wiki | 0-shot$^6$ |
> > | $_\text{(W-A-KV)}$ |  | ($\downarrow$) | Avg.($\uparrow$) | ($\downarrow$) | Avg.($\uparrow$) | ($\downarrow$) | Avg.($\uparrow$) |
> > |  | FP16 | 10.48 | 65.63 | 5.75 | 73.33 | 5.47 | 69.79 |
> > | **4-8-8** | RTN | 40.61 | 47.27 | 77.69 | 45.00 | 72.98 | 47.75 |
> > |  | RTN-opt | 11.20 | 61.09 | 7.32 | 67.35 | 7.11 | 56.93 |
> > |  | QuaRot | 10.89 | 63.12 | 7.04 | 67.60 | 6.22 | 63.43 |
> > |  | SpinQuant | 11.03 | 63.28 | 6.54 | 71.60 | 5.97 | 66.01 |
> > |  | OSTQuant | 11.05 | 62.48 | 6.56 | 71.46 | 6.49 | 61.85 |
> > |  | FlatQuant | 10.67 | 65.04 | 6.20 | 72.11 | 6.46 | 62.07 |
> > |  | FPTQuant | 10.65 | 64.00 | 6.27 | 72.72 | 5.85 | 65.96 |
> > | **4-4-4** | RTN | 2229 | 29.17 | 1.6e5 | 37.67 | 2408 | 39.13 |
> > |  | RTN-opt | 59.06 | 31.16 | 543 | 30.04 | 2220 | 29.54 |
> > |  | QuaRot | 12.81 | 54.38 | 19.72 | 42.76 | 1218 | 30.21 |
> > |  | SpinQuant | 12.71 | 54.88 | 11.04 | 54.58 | 940 | 30.17 |
> > |  | OSTQuant | 13.41 | 52.43 | 9.66 | 56.69 | 519 | 30.75 |
> > |  | FlatQuant | 11.38 | 61.00 | 9.55 | 61.43 | 106 | 29.90 |
> > |  | FPTQuant | 11.71 | 59.46 | 9.74 | 52.96 | 603 | 29.76 |
> >
> > We observe that OSTQuant performs inconsistently on static quantization, with FPTQuant almost always outperforming it. We hypothesize this is due to OSTQuant not being function-preserving and less stable (see sensitivity analysis below), which is more problematic for static quantization than dynamic quantization due to the more difficult learning setting. Since FPTQuant is on average better, and since it has fewer non-mergeable transforms than OSTQuant, it should be the preferred method in all scenarios.
> >
> > *continues*

---

> ### Author Response · Authors · 2025-11-28
>
> We also compare ourselves to OSTQuant on dynamic quantization
>
> **OSTQuant: dynamic quantization (Section 4.4)**
>
> We note that our dynamic results (Table 3) did not use GPTQ, whereas OSTQuant uses this by default. We compare our method with and without GPTQ weight quantization to OSTQuant on L2-7B and L3-8B using W4-A4-KV4 quantization.
>
> We used the provided commands, settings and their code to obtain OSTQuant results.
> For both FPTQuant and OSTQuant we used the default settings of GPTQ & use the same version of lm_eval to ensure consistency.
> We report WikiText2 test perplexity (2048 seq. length) and CSR over 6 tasks used in FPTQuant. We repeated experiments 5 times and reported the mean and standard deviation.
>
> As expected, **OSTQuant performs considerably worse w/o GPTQ**, given that it uses it by default.
> At the same time, we notice that **our method further benefits from GPTQ** (except L2-7B perplexity, which is slightly higher). We almost match the performance of OSTQuant, while using fewer transforms and fewer non-mergeable transforms.
>
> | | L2-7B | | L3-8B |  |
> |---|--|---|---|---|
> | | WT2 @ T=2048 | CSR | WT2 @ T=2048 | CSR |
> | OSTQuant (w/o GPTQ) | 6.37 ± 0.01 | 66.02 ± 0.29 | 7.98 ± 0.01 | 68.32 ± 0.31 |
> | OSTQuant (w/ GPTQ, default) | 5.94 ± 0.01 | 66.61 ± 0.30 | 7.32 ± 0.02 | 68.51 ± 0.33 |
> | FPTQuant (w/o GPTQ, from paper) | 5.97 | 66.06 | 7.64 | 68.11 |
> | FPTQuant (w/ GPTQ) | 6.08 ± 0.01 | 66.31 ± 0.12 | 7.62 ± 0.02 | 68.51 ± 0.50 |
>
>
> > **4.** For static int4 quantization... attention computation typically employs memory-efficient kernels like FlashAttention...
>
> Thank you for highlighting this. **Note that in contrast to all related work (SpinQuant, QuaRot, FlatQuant, OSTQuant), our attention-block transforms are mergeable and thus do not change the attention computation.** This means we can use standard memory-efficient kernels, and do not need to write new kernels that apply online operations.
>
>
> > **0.** I fully acknowledge the paper’s valuable contributions to the community (e.g., residual normalization shows potential for reducing down-projection complexity). However, my evaluation places it slightly below the acceptance threshold, and given the absence of a 5-point scoring option, I can only assign a score of 4.
>
> Though we realize you will not be able to respond to this message anymore, we hope the above resolves the last unclarities. We would like to thank you again for your time and feedback: the sensitivity analysis, new model results, and inclusion of OSTQuant have improved our paper significantly.
>
> ----

---

> > ### Author Response · Authors · 2025-11-28
> > **addendum: sensitivity analysis**
> >
> > # Sensitivity analysis
> >
> > In the earlier comment we showed how OSTQuant performs poorly on static quantization. We would like to explain this.
> >
> > The function-preserving property (desideratum P1) of FPTs is useful because it reduces the capacity to change the pretrained model's output, and consequently can avoid overfitting to calibration data. We have conducted a sensitivity analysis to show the function-preserving properties of different transforms **without quantization**, which also gives insight into training stability.
> >
> > We take the initialized transforms, and simulate noisy training dynamics by perturbing the parameters; we add i.i.d. Gaussian noise with standard deviation $\sigma\in (0.00, 0.1, 0.3, 1.0, 3.0)$ to each transform parameter. Naturally, we ensure the parameterizations remain correct---i.e. that an orthogonal matrix remains orthogonal (using `torch.nn.utils.parameterization`). We run it for three model sizes, of 1B, 3B, and 8B parameters, and for 5 seeds.
> >
> > | L3.2-1B-it | 0.00 | 0.1 | 0.3 | 1.0 | 3.0  |
> > |:----|:----|:-----|:-----|:-----|:-----|
> > | SpinQuant | $13.16^{\pm 0.00}$ | $13.16^{\pm 0.00}$ | $13.16^{\pm 0.00}$ | $13.16^{\pm 0.00}$| $13.16^{\pm 0.00}$  |
> > | FlatQuant | $13.16^{\pm 0.00}$ | $13.16^{\pm 0.00}$ | $13.18^{\pm 0.02}$ | $5.52e+05^{\pm 7.5e+05}$ | $259^{\pm 458.8}$|
> > | OSTQuant | $13.16^{\pm 0.00}$ | $13.20^{\pm 0.02}$ | $19.01^{\pm 4.92}$ | $3.11e+04^{\pm 7.3e+03}$ | $4.08e+04^{\pm 1.5e+04}$ |
> > | FPTQuant | $13.16^{\pm 0.00}$ | $13.16^{\pm 0.00}$ | $13.16^{\pm 0.00}$ | $13.16^{\pm 0.00}$| $13.16^{\pm 0.00}$  |
> >
> > | L3.2-3B-it | 0.00 | 0.1 | 0.3 | 1.0| 3.0  |
> > |:----|:------|:-----|:-----|:-------|:-----|
> > | SpinQuant | $11.05^{\pm 0.00}$ | $11.05^{\pm 0.00}$ | $11.05^{\pm 0.00}$ | $11.05^{\pm 0.00}$| $11.05^{\pm 0.00}$|
> > | FlatQuant | $11.05^{\pm 0.00}$ | $11.05^{\pm 0.00}$ | $11.63^{\pm 1.07}$ | $1.46e+05^{\pm 2.9e+05}$ | $1.30e+05^{\pm 2.5e+05}$ |
> > | OSTQuant | $11.05^{\pm 0.01}$ | $11.06^{\pm 0.05}$ | $14.58^{\pm 1.78}$ | $1.38e+04^{\pm 4.8e+03}$ | $1.38e+04^{\pm 5.4e+03}$ |
> > | FPTQuant | $11.05^{\pm 0.00}$ | $11.05^{\pm 0.00}$ | $11.05^{\pm 0.00}$ | $11.05^{\pm 0.00}$| $11.05^{\pm 0.00}$|
> >
> > | L3-8B | 0.00|0.1 | 0.3 | 1.0| 3.0  |
> > |:----|:-------|:-------|:-------|:--------|:-----|
> > | SpinQuant | $6.14^{\pm 0.00}$ | $6.14^{\pm 0.00}$ | $6.14^{\pm 0.00}$ | $6.14^{\pm 0.00}$ | $6.14^{\pm 0.00}$ |
> > | FlatQuant | $6.14^{\pm 0.00}$ | $6.14^{\pm 0.00}$ | $6.35^{\pm 0.14}$ | $8.83e+05^{\pm 1.1e+06}$ | $4.77e+05^{\pm 5.6e+05}$ |
> > | OSTQuant | $6.14^{\pm 0.00}$ | $6.15^{\pm 0.00}$ | $8.02^{\pm 1.12}$ | $2.67e+04^{\pm 1.5e+04}$ | $3.16e+04^{\pm 1.4e+04}$ |
> > | FPTQuant | $6.14^{\pm 0.00}$ | $6.14^{\pm 0.00}$ | $6.14^{\pm 0.00}$ | $6.14^{\pm 0.00}$ | $6.14^{\pm 0.00}$ |
> >
> > We observe that SpinQuant and FPTQuant are very stable---even when we completely randomize the transform parameters, we are ensured that the function-preserving properties are in fact, preserved, and that output is stable. FlatQuant is theoretically function-preserving, but is less stable: their approach consists of 6 transforms per transformer block, which each have 1 or 2 online matrix inversions. The latter can result in floating point precision issues which destroy the function-preservation. We have found this is not an issue during training as long as a small learning rate is chosen (i.e. the noise is small and the optimizer can correct errors in later steps).
> >
> > OSTQuant is not function-preserving; it uses smoothing transforms that do not cancel each other out. For example, their $S_{qk}$ transform does not commute with RoPE, and hence the query and key transforms do not cancel out (i.e. no function-preservation). We see this in the results. Smoothing vectors are initialized as identities, hence without noise the model works as expected (yields idential output to the original full precision network). When the weights are updated, the function-preservation is quickly lost. This also means that the model has capacity to overfit the training data.

---

> > > ### Author Response · Authors · 2025-12-03
> > > ****Summary for AC****
> > >
> > > - **W1**_,Q1_: Added OSTQuant results (**Tables 2 and 3**). OSTquant is not function-preserving (**Appendix L**). FPTQuant outperforms OSTQuant on static quantization and matches dynamic performance while using fewer transforms.
> > > - **W2:** Clarified speed–accuracy trade-off: FPTQuant is up to 29% faster than FlatQuant, performs consistently better than SpinQuant, and hence offers best Pareto trade-off for deployment scenarios.
> > > - **W3:** Changed colour scheme Figure 1.
> > > - **W4**_,Q2_: Added Qwen 32B results to show scalability (**Table 2**). Clarified FPTQuant trains faster than Flatquant (**Appendix J**) and transform is more stable (**Appendix L**).
> > > - _Q3_: Added GPTQ to **Table 3**.

---

### Official Review · Reviewer_h3nK · 2025-10-31

**Soundness:** 2
**Presentation:** 3
**Contribution:** 2
**Rating:** 4
**Confidence:** 4

**Summary:**

This paper addresses the key challenge of performance degradation in LLM quantization caused by large-magnitude outliers, proposing FPTQuant—a method leveraging function-preserving transforms (FPTs) to balance quantization efficiency and model accuracy. FPTQuant introduces three novel lightweight FPTs (mergeable pre-RoPE for queries/keys, mergeable per-head transform for values, dynamic per-token residual scaling) and integrates existing effective transforms (e.g., Hadamard, rotation). These FPTs exploit transformer equivariances/independencies to reshape activation distributions for better quantization compatibility while preserving model function; crucially, most are mergeable into existing weights, adding nearly no inference overhead and requiring no custom kernels. Empirically, FPTQuant enables static INT4 quantization with up to 3.9× speedup over FP models on NVIDIA RTX 3080 Ti. Across Llama-series models and tasks (Wikitext-2 perplexity, zero-shot CSR, MMLU), it matches or outperforms baselines (QuaRot, SpinQuant) and only slightly lags the 29%-slower FlatQuant—excelling particularly in realistic static quantization scenarios (quantizing more intermediate activations) where prior work falls short. Ablations validate individual FPT contributions, and practical guidelines for FPT selection are provided.

**Strengths:**

1. Minimal Inference Overhead with Mergeable Transforms. Most FPTs (e.g., pre-RoPE for queries/keys, per-head value transform) can be merged into existing model weights, avoiding extra computational cost or custom kernels during inference, which is critical for practical LLM deployment, especially on edge devices
2. Leverages transformer equivariances/independencies and two-stage optimization, local L_p-norm minimization + end-to-end student-teacher training to reshape activation distributions, addressing the core issue of outlier-induced quantization error
3. Enables static INT4 quantization with up to 3.9× speedup over FP models, while matching or outperforming baselines (QuaRot, SpinQuant) on tasks like Wikitext-2 perplexity and zero-shot CSR; it only slightly lags the 29%-slower FlatQuant

**Weaknesses:**

1. For very challenging setups (e.g., W4A4KV4 on Llama 2 7B), FPTQuant’s accuracy gap with FlatQuant widens, especially in zero-shot reasoning—indicating limitations in handling severe quantization pressure
2. While inference is lightweight, FPTQuant’s two-stage optimization (local L_p-norm minimization + end-to-end student-teacher training) adds more training steps than simpler PTQ methods (e.g., RTN-opt); even with mergeable transforms, training larger models (e.g., Llama 2 7B) takes longer than baselines like QuaRot
3. The core theoretical foundation of this paper lies in the concept of Function-Preserving Transformations (FPTs), which are designed to leave model outputs unchanged in the absence of quantization. However, the introduction of a teacher–student training framework disrupts this theoretical closure: in the distillation process, the student model (quantized + equipped with FPTs) is trained to approximate the output distribution of the full-precision teacher model. This fitting process inevitably adjusts the parameters of the FPTs—meaning that even without quantization, these updated FPTs may deviate from their originally function-preserving design. The paper does not provide theoretical or empirical evidence ensuring that the transformations remain equivalent after distillation. Consequently, it remains unclear whether the observed performance improvements stem primarily from the theoretically equivalent transformations or from the subsequent distillation training itself.

**Questions:**

1. The local optimization stage is said to improve end-to-end training stability. However, the paper only provides results for p=4 (following Bondarenko et al., 2024) without ablating other p values (e.g., p=2, p=∞. Why is p=4 optimal for FPTQuant, and how sensitive is the method’s performance to this hyperparameter—especially for models with different activation distributions?
2. While FPTQuant is tested on Llama-series models and Qwen 2.5 7B, it lacks evaluation on LLMs with distinct architectures (e.g., Mistral with sliding window attention, GPT-4 derivatives) or non-standard activation functions. Given that outlier patterns vary across architectures (as seen in Qwen’s degraded performance), how can the authors claim FPTQuant’s broader applicability without validating it on more diverse model families?
3. Can a statistical analysis be conducted on the overall time-consuming costs of the FPTQuant method, including both the quantization and distillation training processes, and can the convergence criteria be provided? What advantages does it have over other quantization methods such as FlatQuant? Or can it achieve better convergence performance?

---

> ### Author Response · Authors · 2025-11-21
> **Rebuttal [1/n]**
>
> Many thanks for your positive and constructive feedback! We especially appreciate your recognition of how FPTQuant's mergeability is critical for fast, acurrate, and practical LLM deployment.
>
> ### Weaknesses
>
> **1. Accuracy gap FlatQuant**
> > For very challenging setups (e.g., W4A4KV4 on Llama 2 7B), FPTQuant’s accuracy gap with FlatQuant widens, especially in zero-shot reasoning—indicating limitations in handling severe quantization pressure
>
> We completely understand your concern. There is a speed-accuracy trade-off (see Section 5 Discussion, paragraph "FPTQuant"). We believe FPTQuant achieves a better trade-off compared to baselines:
> 1. FPTQuant outperforms all baselines (incl. FlatQuant) on the most challenging setting, W4A4KV4 with all activations quantized (Table 1, right column)
> 2. FPTQuant outperforms SpinQuant almost consistently (Table 2), while being comparable in speed (Figure 2). SpinQuant also has an additional unmergeable transform (after RoPE) that we do not have.
> 3. FPTQuant indeed does slightly worse than FlatQuant for the setting they report in their paper, `Linears+KV` (see Table 2, bottom). However, FPTQuant is up to 29\% faster due to the many online transforms of FlatQuant, hence FPTQuant still provides a favourable speed-accuracy trade-off.
> 4. When more performance is needed, non-mergeable transforms can be added to FPTQuant to change the speed-accuracy trade-off (see Appendix K). In Table 12, we added $P_h$ (from FlatQuant) to FPTQuant—a full "cross-channel" QK transform—which for W4A4KV4 Llama 3.2 3B-it narrows the gap to FlatQuant from +0.31 PPL to +0.16 PPL, and CSR from +1.63 to just +0.39.
>
> Overall, we believe FPTQuant gives in most cases the best accuracy-efficiency trade-off, but for cases in which a different operating point is desired (e.g. higher accuracy), FPTQuant can be combined with other existing methods to yield the most accurate model given the performance constraints.
>
> **2. Training cost**
> > While inference is lightweight, FPTQuant’s two-stage optimization (local L_p-norm minimization + end-to-end student-teacher training) adds more training steps than simpler PTQ methods (e.g., RTN-opt); even with mergeable transforms, training larger models (e.g., Llama 2 7B) takes longer than baselines like QuaRot.
>
> *Local optimization stage.* We note that the local optimization should be intepreted as a good initialization step; it is negligible in total cost (<1% of end-to-end training, see Appendix J.1), and has no hyperparameters (besides loss function).
>
> *End-to-end training stage.* Note that all methods, including RTN and QuaRot, need training of the quantization grid to ensure good performance for static quantization at low bitwidths. We included some results for this in Table 2, but let us now also include QuaRot with (-opt) and without optimization of quantization grid parameters:
>
> **Table [optimization of baselines]**: RTN and QuaRot with (-opt) and without optimization of the quantization grid. Without optimization, RTN and QuaRot are completely broken at W4A4KV4, and perform poorly even at W4A8KV8. Optimization is thus essential for low bitwidth static quantization, even for baselines. *N.B. The original QuaRot paper does not optimize the quantization grid, but for fair comparison, we do do this in our paper. Consequently, the results here denoted by "QuaRot-opt" are the results denoted as "QuaRot" in our paper.*
>
> |  |  | L3.2 3B-it |  | L3 8B |  |
> |:---:|---|:---:|:---:|:---:|:---:|
> | \#Bits | Method | Wiki | 0-shot$^6$ | Wiki | 0-shot$^6$ |
> | $_\text{(W-A-KV)}$ |  | ($\downarrow$) | Avg.($\uparrow$) | ($\downarrow$) | Avg.($\uparrow$) |
> |   | FP16 | 10.48 | 65.63 | 5.75 | 73.33 |
> | 4-8-8 | RTN | 40.61 | 47.27 | 77.69 | 45.00 |
> |  | RTN-opt | 11.20 | 61.09 | 7.32 | 67.35 |
> |  | QuaRot | 20.93 | 52.93 | 39.85 | 47.05 |
> |  | QuaRot-opt | 10.89 | 63.12 | 7.04 | 67.60 |
> | 4-8-4 | RTN | 127.9 | 40.40 | 126.9 | 41.46 |
> |  | RTN-opt | 11.57 | 58.92 | 7.78 | 64.73 |
> |  | QuaRot | 56.76 | 36.29 | 77.28 | 32.83 |
> |  | QuaRot-opt | 11.09 | 63.18 | 7.29 | 66.71 |
> | 4-4-4 | RTN | 2229 | 29.17 | 1.6e5 | 37.67 |
> |  | RTN-opt | 59.06 | 31.16 | 543 | 30.04 |
> |  | QuaRot | 1.0e5 | 29.79 | 9.7e5 | 29.86 |
> |  | QuaRot-opt | 12.81 | 54.38 | 19.72 | 42.76 |
>
> **_continues_**

---

> ### Author Response · Authors · 2025-11-21
> **Rebuttal [2/3]**
>
> We see that training (-opt) of the quantization grid is essential.
>
> Still, you are right: FPTQuant does need more training time than simpler baselines, e.g. QuaRot-opt is -25% as fast on L3.2 3B-it (see Appendix, Table 18). SpinQuant is sometimes faster, sometimes slower than FPTQuant. FlatQuant is consistently slower (e.g. +45% on Llama 3.2 3B-it).
> Since training is a one-time cost and still manageable (<1 A100 day for all models), we think that in most settings this extra cost will weigh up against the additional performance and speed gains at infererence time. It is a large plus that the training cost of our transforms is lower than that of our closest competitor, FlatQuant (see also Question 3). *Sidenote: the mergeability also yields a simpler inference set-up compared to baselines QuaRot/SpinQuant/FlatQuant; most FPTQuant transforms are merged so deploying new models to different devices efficiently is easier.*
>
> **3. FPTs not preserving after training?**
> > The core theoretical foundation of this paper lies in the concept of Function-Preserving Transformations (FPTs), which are designed to leave model outputs unchanged in the absence of quantization. However, the introduction of a teacher–student training framework disrupts this theoretical closure: in the distillation process, the student model (quantized + equipped with FPTs) is trained to approximate the output distribution of the full-precision teacher model. This fitting process inevitably adjusts the parameters of the FPTs—meaning that even without quantization, these updated FPTs may deviate from their originally function-preserving design. The paper does not provide theoretical or empirical evidence ensuring that the transformations remain equivalent after distillation. Consequently, it remains unclear whether the observed performance improvements stem primarily from the theoretically equivalent transformations or from the subsequent distillation training itself.
>
> Thank you for highlighting this concern. Most importantly, **the weights of the original model are frozen** (see Figure 1). Consequently, the "student" is almost identical to the teacher, besides the added quantizers and transforms (the parameters of which are learnt, see L.282). This is also why we do not describe it as *distillation* (as you write), but only *training of transforms and quantization grids*. To avoid confusion, we **have emphasized in Section 3.2.2 that the original weights remain frozen**.
>
> Secondly, The FPTs are parameterized such that they remain FPTs during training. In particular, $T_r$ is parameterized as orthogonal, $T_k$ as a 2x2 scaled-rotation, $T_v$ as an invertible matrix, $T_u$ as a scaling vector, and the ``inverses'' for each of these (denoted by bars) are computed on-the-fly to ensure that the pair (e.g. $T_k$ and $\bar{T}_k$) remain FPT. Consequently, whatever is learnt during training, **without quantization, the output of the network with FPTs never changes**. As a result, the improvements really are due to the transforms, not to the training. Does this alleviate your concern?
>
> ### Questions
>
> **1. Choosing $p$ for $L_p$ local optimization**
> > The local optimization stage is said to improve end-to-end training stability. However, the paper only provides results for p=4 (following Bondarenko et al., 2024) without ablating other p values (e.g., p=2, p=∞. Why is p=4 optimal for FPTQuant, and how sensitive is the method’s performance to this hyperparameter—especially for models with different activation distributions?
>
> **We have added new results to Appendix F.2.1 that studies different $p$, see Table 14.** For visibility, we include the description here:
> > We ablate different values of $p$ for the same setting. We find that no $p$ performs significantly better than another—though not using local optimization ("No opt") does significantly worse on average and has a large variance. As before (Figure 5), this shows that local optimization improves training stability, though the choice of $p$ is perhaps less important.

---

> ### Author Response · Authors · 2025-11-21
> **Rebuttal [3/3]**
>
> **2. More models**
> > While FPTQuant is tested on Llama-series models and Qwen 2.5 7B, it lacks evaluation on LLMs with distinct architectures (e.g., Mistral with sliding window attention, GPT-4 derivatives) or non-standard activation functions. Given that outlier patterns vary across architectures (as seen in Qwen’s degraded performance), how can the authors claim FPTQuant’s broader applicability without validating it on more diverse model families?
>
> Thank you for raising this concern. It would indeed be interesting to see how different model families may behave differently. Unfortunately, we had a limited compute and time budget (e.g. GPT-OSS was only released in August), hence we mostly followed the most related work (QuaRot, SpinQuant and FlatQuant), who evaluated Llama models exclusively. We hope the reviewer appreciates that we added Qwen—a model that seems harder, also for FPTQuant—, that we covered models of different sizes, that we added additional quantization settings (Section 4.2), additional metrics (Table 16), and have ablations in Appendix F that can give insight into the utility of transforms for other models (e.g. models that have a different attention mechanism may still benefit from similar behaviour in the FFN).
>
> *Recent models are all similar in architecture.* We also note that most recent architectures are very similar: recent LLMs (Mistral 3.x, Llama, Qwen, Gwen, GPT-OSS) use Swish as activation function. They effectively have stopped using sliding window attention: Mistral 3 no longer uses sliding window (e.g. see `https://huggingface.co/mistralai/Mistral-Small-3.1-24B-Instruct-2503/blob/main/config.json`), and even "Ministral" uses window sizes of 32k+, which is larger than the perplexity or zero-shot context length (see `https://huggingface.co/mistralai/Ministral-8B-Instruct-2410/blob/main/config.json`).
>
> To make the paper even more relevant to future readers, we **are adding results for Mistral's Ministral 8B instruct.**
>
> **_EDIT: Table 2 now includes results for Ministral 8B-it and Qwen 2.5 32B_**
>
> **3. Convergence criteria/training time**
> > Can a statistical analysis be conducted on the overall time-consuming costs of the FPTQuant method, including both the quantization and distillation training processes, and can the convergence criteria be provided? What advantages does it have over other quantization methods such as FlatQuant? Or can it achieve better convergence performance?
>
> *End-to-end training.*
> We do not do distillation (see also Weakness 3), we just train FPTs/quantization parameters, and the total training time is favorable compared to baselines (see Appendix J, also our response to Weakness 2). It makes sense FPTQuant's training is favorable, when we analyse the student-teacher training and compare it to other training set-ups:
> 1. End-to-end student-teacher training yields better generalization performance than SpinQuant's next-token prediction, because it provides more signal and is less prone to overfitting the training dataset (as we show empirically in Appendix F.2.2).
> 2. Whether we do end-to-end training or training per block (FlatQuant), each update requires a full forward and backwards pass through each block. Training per block may even require forward passes through preceding layers to get the input for block $i$ (and caching of block $i$'s inputs becomes intractable for large datasets).
> 3. Whether we use student-teacher training or next-token prediction does not matter significantly for training cost. The teacher model has zero memory footprint, since it shares its weights with the student. Getting the teacher logits is also marginal: the teacher is run without gradient, backwards pass, and without transforms (which for the student cause the most significant overhead during training, e.g. see Table 18).
> 4. FPTQuant is significantly faster than FlatQuant, because the FlatQuant transforms (Kronecker decompositions of invertible matrices) are expensive to train. The most expensive FPTQuant transforms is the single rotation $T_r$ (which requires one expensive parameterization, see Table 18). SpinQuant has a similar cost, since it also uses $T_r$. FlatQuant on the other hand, requires online matrix inverses for each of its six transforms *per block*. This explains why we see a training time that is up to 45% slower for FlatQuant.
>
> *Local optimization.* The local optimization (e.g. Eq. 10) is a convex problem, which converges fast—i.e. the overhead of this "initialization step" is just a few minutes for the whole model (<1% training time). More importantly, we have found (**Appendix F.2.1**) that the local optimization *helps* the convergence of the subsequent end-to-end training. In particular, it provides a good "initialization" (Table 13), leading to more stable subsequent end-to-end training (Figure 5).
>
> **_We have updated the PDF, with changes highlighted in green_**

---

> > ### Comment · Reviewer_h3nK · 2025-11-25
> >
> > Although the paper presents a promising approach (FPTQuant) that incorporates several simple, integrable lightweight transforms to improve static quantization, the current version still has many unresolved critical issues—particularly structural ones—that limit the paper’s scientific rigor and completeness.
> >
> > The theoretical basis for "function-preserving transforms" is insufficiently validated: After integrating these transforms into the model and performing end-to-end training, it remains unclear whether they can still strictly maintain "functional equivalence"—especially for those dense, unconstrained value-transform matrices. The paper provides neither mathematical proofs nor experimental evidence to demonstrate that the original function’s behavior is preserved post-training.
> >
> > The pseudo-dynamic residual scaling mechanism carries potential numerical instability: Due to its recursive scaling nature, this mechanism may cause numerical fluctuations. However, the paper fails to specify the theoretical bounds of this instability, conduct sensitivity analysis (i.e., how it responds to parameter changes), or investigate its stability in deep networks. This omission makes it difficult to assess the method’s robustness in practical deployments.
> >
> > The rationale for the local optimization strategy lacks sufficient support: The strategy overrelies on L₄ minimization but does not conduct adequate comparative experiments (e.g., testing other norm choices) or justify why L₄ is the optimal selection. This weakens the logical clarity of the proposed optimization pipeline.
> >
> > Key comparative experiments are missing, and performance is unstable in extreme scenarios: The paper does not compare with OSTQuant—the most directly relevant method—significantly undermining the credibility of its experimental conclusions. Additionally, the method exhibits substantial performance volatility in extreme quantization settings (e.g., W4A4KV4), with a drastically widened performance gap compared to FlatQuant.
> >
> > In summary, while the approach is innovative and promising, the aforementioned issues indicate that the submission lacks a solid theoretical foundation and comprehensive experimental validation required for a robust assessment. Given the current state of the work, the existing scores are appropriate, and maintaining the current evaluation is reasonable.

---

> > > ### Author Response · Authors · 2025-11-28
> > >
> > > Many thanks for your feedback! Let us respond to your different points
> > >
> > > > **1.** The theoretical basis for "function-preserving transforms" is insufficiently validated: After integrating these transforms into the model and performing end-to-end training, it remains unclear whether they can still strictly maintain "functional equivalence"—especially for those dense, unconstrained value-transform matrices. The paper provides neither mathematical proofs nor experimental evidence to demonstrate that the original function’s behavior is preserved post-training.
> > >
> > > **Theoretical evidence**
> > > Thank you for raising this concern. We want to stress the function-preserving nature of each transform. Theorem 3.1 proves $\mathbf{T}_k$ is function-preserving. $\mathbf{T}_u$ is function-preserving, since for an entry-wise product $\mathbf{A}\odot\mathbf{B}$ and diagonal matrix $\mathbf{T}^{-1}_u$, it holds $(\mathbf{A}\odot\mathbf{B})\mathbf{T}^{-1}_u=(\mathbf{A}\mathbf{T}^{-1}_u)\odot\mathbf{B}$, hence we can merge the inverse into the down projection. Section 3.1.3 proves $\mathbf{S}_n$ is function-preserving, where the most important step uses that for any linear layer $f(\mathbf{X})=\mathbf{XA}$, and (representing scaling vectors as diagonal) $\mathbf{S}_n$, it holds: $\mathbf{S}_n f(\mathbf{X})v=f(\mathbf{S}_n\mathbf{X})$, i.e. we can put the dynamic scaling inside the blocks (line 253).
> > >
> > > The function-preserving property of the value transform relies on the linearity of batch-matrix-multiplications. Let us prove this in detail. The attention mechanism mixes across tokens, whereas the value and out projection matrices are applied to the hidden dimension (see the excellent work by) [(Elhage et al, 2021)](https://transformer-circuits.pub/2021/framework/index.html)), i.e. for attention weights $A$, linear layers $W_v, W_o$, and input $X$, the output of the attention block is $(\mathbf{A} (\mathbf{X}\mathbf{W}_v))\mathbf{W}_o$. Applying $T_v$ and $T^{-1}_v$ as we describe in Section 3.1.2, we get $(\mathbf{A} (\mathbf{X}[\mathbf{W}_v T_v])[T_v^{-1}\mathbf{W}_o]=(\mathbf{A} (\mathbf{X}\mathbf{W}_v))\mathbf{W}_o$, i.e. the output of the attention block remains the same. Note that $T_v^{-1}$ exists with probability 1, since non-invertible matrices form a null space under stochastic optimization. Grouped query attention is handled by using a different transform per head, see Eq. 6. We agree that our original Section 3.1.2 could have been more explicit---**we will include this in Section 3.1.2**.
> > >
> > > We hope this alleviates your concern. We also note that we have unit tests that disable quantization, and assert that FPTQuant transforms do not change the output of the whole network even after optimization (besides absolute/relative error on the order of `1e-4`, to account for FP precision error).
> > >
> > > **Experimental evidence.** We also provide experimental evidence of function-preservation in the next point's, "Sensitivity analysis".
> > >
> > > > **2.** The pseudo-dynamic residual scaling mechanism carries potential numerical instability: Due to its recursive scaling nature, this mechanism may cause numerical fluctuations. However, the paper fails to specify the theoretical bounds of this instability, conduct sensitivity analysis (i.e., how it responds to parameter changes), or investigate its stability in deep networks. This omission makes it difficult to assess the method’s robustness in practical deployments.
> > >
> > > We understand the concern, and thank you for the suggestion of doing a sensitivity analysis! We include this here:
> > >
> > > **Sensitivity analysis**
> > >
> > > We perturb transform weights with random updates to test stability, and function-preservation. We show empirically that **FPTQuant remains function-preserving during optimization.** Some baselines do not.
> > >
> > > We take the initialized transforms, and simulate noisy training dynamics by perturbing the parameters; we add i.i.d. Gaussian noise with standard deviation $\sigma\in \{0.00, 0.1, 0.3, 1.0, 3.0\}$ to each transform parameter. Naturally, we ensure the parameterizations remain correct---i.e. that an orthogonal matrix remains orthogonal (using `torch.nn.utils.parameterization`). We run it for three model sizes, of 1B, 3B, and 8B parameters, and for 5 seeds.
> > >
> > > *...continues*

---

> > > > ### Author Response · Authors · 2025-11-28
> > > >
> > > > *...continued sensitivity analysis*
> > > >
> > > > | L3.2-1B-it | 0.00 | 0.1 | 0.3 | 1.0 | 3.0  |
> > > > |:----|:----|:-----|:-----|:-----|:-----|
> > > > | SpinQuant | $13.16^{\pm 0.00}$ | $13.16^{\pm 0.00}$ | $13.16^{\pm 0.00}$ | $13.16^{\pm 0.00}$| $13.16^{\pm 0.00}$  |
> > > > | FlatQuant | $13.16^{\pm 0.00}$ | $13.16^{\pm 0.00}$ | $13.18^{\pm 0.02}$ | $5.52e+05^{\pm 7.5e+05}$ | $259^{\pm 458.8}$|
> > > > | OSTQuant | $13.16^{\pm 0.00}$ | $13.20^{\pm 0.02}$ | $19.01^{\pm 4.92}$ | $3.11e+04^{\pm 7.3e+03}$ | $4.08e+04^{\pm 1.5e+04}$ |
> > > > | FPTQuant | $13.16^{\pm 0.00}$ | $13.16^{\pm 0.00}$ | $13.16^{\pm 0.00}$ | $13.16^{\pm 0.00}$| $13.16^{\pm 0.00}$  |
> > > >
> > > > | L3.2-3B-it | 0.00 | 0.1 | 0.3 | 1.0| 3.0  |
> > > > |:----|:------|:-----|:-----|:-------|:-----|
> > > > | SpinQuant | $11.05^{\pm 0.00}$ | $11.05^{\pm 0.00}$ | $11.05^{\pm 0.00}$ | $11.05^{\pm 0.00}$| $11.05^{\pm 0.00}$|
> > > > | FlatQuant | $11.05^{\pm 0.00}$ | $11.05^{\pm 0.00}$ | $11.63^{\pm 1.07}$ | $1.46e+05^{\pm 2.9e+05}$ | $1.30e+05^{\pm 2.5e+05}$ |
> > > > | OSTQuant | $11.05^{\pm 0.01}$ | $11.06^{\pm 0.05}$ | $14.58^{\pm 1.78}$ | $1.38e+04^{\pm 4.8e+03}$ | $1.38e+04^{\pm 5.4e+03}$ |
> > > > | FPTQuant | $11.05^{\pm 0.00}$ | $11.05^{\pm 0.00}$ | $11.05^{\pm 0.00}$ | $11.05^{\pm 0.00}$| $11.05^{\pm 0.00}$|
> > > >
> > > > | L3-8B | 0.00|0.1 | 0.3 | 1.0| 3.0  |
> > > > |:----|:-------|:-------|:-------|:--------|:-----|
> > > > | SpinQuant | $6.14^{\pm 0.00}$ | $6.14^{\pm 0.00}$ | $6.14^{\pm 0.00}$ | $6.14^{\pm 0.00}$ | $6.14^{\pm 0.00}$ |
> > > > | FlatQuant | $6.14^{\pm 0.00}$ | $6.14^{\pm 0.00}$ | $6.35^{\pm 0.14}$ | $8.83e+05^{\pm 1.1e+06}$ | $4.77e+05^{\pm 5.6e+05}$ |
> > > > | OSTQuant | $6.14^{\pm 0.00}$ | $6.15^{\pm 0.00}$ | $8.02^{\pm 1.12}$ | $2.67e+04^{\pm 1.5e+04}$ | $3.16e+04^{\pm 1.4e+04}$ |
> > > > | FPTQuant | $6.14^{\pm 0.00}$ | $6.14^{\pm 0.00}$ | $6.14^{\pm 0.00}$ | $6.14^{\pm 0.00}$ | $6.14^{\pm 0.00}$ |
> > > >
> > > > We observe that SpinQuant and FPTQuant are very stable---even when we completely randomize the transform parameters, we are ensured that the function-preserving properties are in fact, preserved, and that output is stable. FlatQuant is theoretically function-preserving, but is less stable: their approach consists of 6 transforms per transformer block, which each have 1 or 2 online matrix inversions. The latter can result in floating point precision issues which destroy the function-preservation. We have found this is not an issue during training as long as a small learning rate is chosen (i.e. the noise is small and the optimizer can correct errors in later steps).
> > > >
> > > > OSTQuant is not function-preserving; it uses smoothing transforms that do not cancel each other out. For example, their $S_{qk}$ transform does not commute with RoPE, and hence the query and key transforms do not cancel out (i.e. no function-preservation). We see this in the results. Smoothing vectors are initialized as identities, hence without noise the model works as expected (yields idential output to the original full precision network). When the weights are updated, the function-preservation is quickly lost. This also means that the model has capacity to overfit the training data.
> > > >
> > > >
> > > > *Sidenote: we note that the stability of the pseudo-dynamic residual scaler ($\mathbf{S}_n$) is good because the the norm between layers changes only sparingly---if a token norm becomes large, it will remain large. This in turn implies that the $\mathbf{S}_n$ will be close to $1$ for most tokens/layers. As shown above empirically, we note that FPTQuant (which in all cases uses dynamic scaling) remains function-preserving.*
> > > >
> > > > > **3.** The rationale for the local optimization strategy lacks sufficient support: The strategy overrelies on L₄ minimization but does not conduct adequate comparative experiments (e.g., testing other norm choices) or justify why L₄ is the optimal selection. This weakens the logical clarity of the proposed optimization pipeline.
> > > >
> > > > In **Appendix F.2.1 we showed that local optimization yields a good initialization for transforms**, since it improves the subsequent end-to-end (E2E) training stability (Figure 5). We do not claim that $L_4$ is the best. We have included a new table (Table 14) that explores different $L_p$ norms. After training, no $p\in (3,4,5,6)$ performs significantly better, but all $L_p$ perform significantly better than no optimization.
> > > >
> > > > > **4.** Key comparative experiments are missing... FlatQuant.
> > > >
> > > > As we showed above empirically, **OSTQuant is not function-preserving** and is thus not a set of FPTs. We agree however that the paper would benefit from a comparison, and have **updated the paper to include OSTQuant results** (please see response to reviewer Hwow below for results). We show FPTQuant outperforms OSTQuant on static quantization and performs comparable on dynamic quantization (better without GPTQ). FPTQuant is also theoretically faster, since have fewer transforms and fewer non-mergeable transforms, and easier to implement (e.g. we can use standard optimized attention kernels, whereas all baselines require online query-key transforms that would need custom kernels).

---

> > > > > ### Author Response · Authors · 2025-11-28
> > > > >
> > > > > ***4.** continued*
> > > > >
> > > > > Finally, we understand that some people will prefer FlatQuant over FPTQuant, and that is totally fine. There are probably also many that might prefer full-network QAT over FlatQuant or FPTQuant. For quantization, or more generally inference efficiency, we always have a trade-off between accuracy, efficiency and effort. We believe it is essential for the community to have a diverse set of techniques with varying characteristics, as the relative importance of different KPIs—such as accuracy, efficiency, or implementation/training effort—depends heavily on the specific use case. In our paper we showed that **FPTQuant has a very favorable accuracy-speed trade-off**, so we hope you agree it is a valuable addition to the techniques available for partitioners.
> > > > >
> > > > > ___
> > > > >
> > > > > Though we realize you will not be able to respond to this message anymore, we hope the above resolves the last unclarities. We would like to thank you again: the sensitivity analysis, new model results, and inclusion of OSTQuant have improved our paper significantly.

---

> > > > > > ### Author Response · Authors · 2025-12-03
> > > > > > ****Summary for AC****
> > > > > >
> > > > > > - **W1:** Highlighted/extended formal proofs for all transforms (**Section 3.1**) and clarified parameterization guarantees. Added sensitivity analysis confirming function-preservation post-training (**Appendix L**).
> > > > > > - **W2:** Conducted numerical stability analysis for pseudo-dynamic scaling; FPTQuant results show robustness across perturbation scales and model sizes (**Appendix L**).
> > > > > > - **W3**_,Q1_: Ablated different norms for local optimization (**Appendix F.2.1**); confirmed L₄ is not critical but local optimization improves convergence stability.
> > > > > > - **W4:** Clarified why FPTQuant is very scalable, and addded larger model Qwen 32B to show this (**Table 2**)
> > > > > > - _Q2_: Besides Qwen, added Ministral 8B to **Table 2** to demonstrate FPTQuant benefit across more model classes.
> > > > > > - _Q3_: Clarified FPTQuant is cheaper, and more stable (due to W1/W2/W3)

---

### Official Review · Reviewer_9RjL · 2025-10-31

**Soundness:** 3
**Presentation:** 2
**Contribution:** 3
**Rating:** 4
**Confidence:** 4

**Summary:**

This paper proposes a framework called FPTQuant, which improves static quantization in Large Language Models (LLMs) by introducing Function-Preserving Transforms (FPTs).

The main innovations of the paper include:

Pre-RoPE Transform: A mergeable 2×2 scaling-rotation matrix pair is inserted before RoPE, allowing Q/K projections to be quantized while preserving the attention output.

Pseudo-Dynamic Residual Scaling: The scaling coefficients of RMSNorm are extended to the residual path to balance the magnitude differences between different tokens, thereby reducing the discreteness of the activation distribution.

Multi-Head Value Transform: An invertible matrix transformation is introduced along the value vector dimension, allowing each head to be trained independently and directly incorporated into the weights.

This method achieves near-state-of-the-art (SOTA) accuracy with static INT4 quantization, while boosting inference speed by up to 3.9x, without requiring custom operators or kernel modifications. Experiments show that its performance outperforms SpinQuant and QuaRot, and is close to FlatQuant in accuracy.

**Strengths:**

1. The pre-RoPE transformation proposed by the authors elegantly solves the merging problem of RoPE in quantization, achieving "functional equivalence" through-RoPE, a first in existing methods.

2. Applying RMSNorm to the residual path equivalently introduces an approximate dynamic normalization mechanism, which is beneficial to quantization stability.

3. A comprehensive comparison was conducted on models such as LLaMA 2, LLaMA 3, and 3B instruct, covering various quantization bit widths and static/dynamic schemes. The results consistently outperform or approach mainstream methods.

4. Compared to the dynamic quantization of SpinQuant/FlatQuant (which requires calculating quantization parameters token by token), FPTQuant achieves good accuracy under completely static configuration (outperforming QuaRot and approaching FlatQuant on W4A8KV8). This is very practical for hardware deployment (NPU/Edge).

**Weaknesses:**

1. Lack of theoretical guarantees for optimization: The rotation matrix of Pre-RoPE is obtained only through local optimization by minimizing the L4 norm; the paper does not provide convergence or global optimality analysis.

2. Lack of distribution validation for residual scaling: Although the authors claim that this mechanism can reduce the amplitude difference between tokens, the paper does not provide activation distribution or outlier visualization, making it difficult to intuitively understand its actual effect.

3. Risk of overfitting when using scale to train quantization coefficients statically: As shown in Table 2, PPL shows significant improvement on Wikitext, but the zero-shot accuracy decreases even more, indicating that scale/zero-point may overfit to the calibration set.

4. Lack of direct comparison with OSTQuant: OSTQuant also uses mergeable rotation and scaling transformations and jointly optimizes their proportion and orthogonality to match the quantized distribution. FPTQuant, as an extension of the same idea, does not conduct direct comparisons, making its relative advantages unclear.

[1]. OstQuant: Refining Large Language Model Quantization with Orthogonal and Scaling Transformations for Better Distribution Fitting

**Questions:**

Please see weaknesses.

---

> ### Author Response · Authors · 2025-11-21
>
> Thank you for highlighting the strengths of FPTQuant's mergeable transforms, the extensive results, good contribution, and importance of static quantization for deployment.
>
> ### Weaknesses
>
> **1. Lack of theoretical guarantees for optimization**
> > The rotation matrix of Pre-RoPE is obtained only through local optimization by minimizing the L4 norm; the paper does not provide convergence or global optimality analysis.
>
> Thank you for highlighting this concern. The local optimization stage converges fast due to the convex loss (Eq. 10), but importantly, we only use this as an initialization and do not assume it gives global optimality. **All transforms** with learneable parameters (i.e. all but $\mathbf{T}_d$ and $\mathbf{S}_n$) **are trained globally** (Eq. 11). This includes the parameters of Pre-RoPE (Eq. 1).
>
> **2. Lack of distribution validation for residual scaling**
> > Although the authors claim that this mechanism can reduce the amplitude difference between tokens, the paper does not provide activation distribution or outlier visualization, making it difficult to intuitively understand its actual effect.
>
> We agree that the paper would benefit from a more intuitive explanation of pseudodynamic residual scaling. We have added intuition in the main paper, Section 3.1.3, Figure **2**. This figure shows (left) that some tokens have serious outliers. These outliers end up in the residual stream, and often lead to large outliers in the next FFN block too (middle). Although we cannot preempt the initial outliers, the dynamic scaling does mean we can reduce the second layer's outliers by taking the norm of the residual into account (right).
>
> **3. Risk of overfitting when using scale to train quantization coefficients statically**
> > As shown in Table 2, PPL shows significant improvement on Wikitext, but the zero-shot accuracy decreases even more, indicating that scale/zero-point may overfit to the calibration set.
>
> We are unsure what you refer to with "the zero-shot accuracy decreases even more". To study the overfitting capability of training the zero-point/scale of static quantization parameters, let us look at Table 2's RTN vs RTN-opt (where the former does not train these parameters, the latter does). We indeed see dramatic reductions in PPL when training. At the same time, the zero-shot performance improves consistently for all methods. *The only two exceptions are the Llama 3 8B and Llama 2 7B models at W4A4KV4, which are completely broken before and after training—optimization does not resolve this.*
> Consequently, there is no evidence of overfitting Wikitext in this table.
>
> *Sidenote: we choose the student-teacher set-up to **avoid** overfitting. The training procedure of SpinQuant, standard next-token prediction, does lead to significant overfitting, as transformations+quantization parameters gives enough capacity for the model to match the next token predictions to the Wikitext vocabulary. See **Appendix F.2.2** for results and a discussion.*
>
> **4. Lack of direct comparison with OSTQuant**
> > OSTQuant also uses mergeable rotation and scaling transformations and jointly optimizes their proportion and orthogonality to match the quantized distribution. FPTQuant, as an extension of the same idea, does not conduct direct comparisons, making its relative advantages unclear.
>
> Thank for you for highlighting this concern. OSTQuant extends SpinQuant. They add a scaling vector to the SpinQuant rotations and add one scale+transform to each value head, which together make their transforms more expressive than SpinQuant's. They also add a Top-K end-to-end loss. OSTQuant is not strictly function-preserving, e.g. their Pre-RoPE scaling $S_{qk}$ does not commute with RoPE and therefore these do not cancel out (i.e. they do not satisfy our desideratum P1).
>
> We hope our paper is valuable to the community in different ways. First, we formalize the idea of function-preserving transforms, including through our proposal of desiderata (P1-P3). This has lead us to design our FPTs based on an analysis of standard transformer operations and their commutation properties—if we understand what operations commute with existing operations, we can improve mergeability _and_ choose the operation with maximal expressivity (e.g. Pre-RoPE as 2x2 scaler+rotations). We hope this systematic approach is valuable for the ICLR community, as it can provide concrete guidance for designing new FPTs that accommodate next-generation transformers with different operations.
>
> *Comparing against OSTQuant.* We have added results for OSTQuant in Table 2 and 3. We find that we outperform OSTQuant on static quantization, since OSTQuant is less stable due to its non-function-preserving transforms (which we show in **Appendix L**). On dynamic quantization we perform comparable. OSTQuant is slower at inference, since it uses online Hadamard for query-key, and smoothing vectors after RMSNorms.

---

> > ### Author Response · Authors · 2025-12-03
> > ****Summary for AC****
> >
> > - **W1:** Clarified that Pre-RoPE transform is trained end-to-end (not just locally). Added sensitivity analysis showing FPTQuant’s stability under perturbations (**Appendix L**).
> > - **W2:** Added **Figure 2** visualizing residual scaling’s effect on outlier mitigation and expanded Section 3.1.3 for intuition.
> > - **W3:** Addressed overfitting concern: explained student–teacher setup avoids overfitting and showed zero-shot accuracy improves consistently (**Appendix F.2.2**).
> > - **W4:** Added OSTQuant comparison in **Tables 2 and 3**; explained OSTQuant is not strictly function-preserving and less stable under static quantization (**Appendix L**).

---

### Comment · Area_Chair_RkZo · 2025-11-24
**Reviewer & Author Discussion**

Hi Reviewers,

Please kindly and actively participate in the review-author discussion if you haven't already, raise your further concerns so that the authors can explain more, and make your final decisions.

Best,
AC

---

### Author Response · Authors · 2025-12-03
**Summary for Area Chair**

Dear Area Chair,

Thank you for your time and efforts in these unusual circumstances. We have uploaded a fully revised manuscript with all requested changes marked in **green**. Below is a summary of the key strengths, the key critiques, and how we addressed them. _We have included summaries per reviewer after each individual reviewer discussion, see `**Summary for AC**`_

---

## **Positive Highlights from Reviews**
- **Novelty and Impact:** Reviewers (9RjL, h3nK, GL33) praised our introduction of *function-preserving transforms* (FPTs), mergeable pre-RoPE query/key transforms, expressive value transforms, and pseudo-dynamic residual scaling—that enable static INT4 quantization without custom kernels and negligible inference overhead. Reviewer Hwow "fully acknowledge[s] the paper’s valuable contributions to the community" through "ingenious adjustments leveraging [standard transformer's] structural characteristics".
- **Strong Empirical Results:** FPTQuant achieves up to **3.9× speedup** while matching or outperforming SpinQuant and QuaRot and approaching FlatQuant (9RjL, h3nK). Results span LLaMA 2/3 and Qwen models across WikiText-2, zero-shot CSR, and MMLU (GL33, 9RjL). Reviewers (Hwow) appreciated detailed ablation studies.
- **Critical for transformer deployment:** Reviewers (9RjL, h3nK, GL33) highlight our ability to fuse transforms into existing weights as critical for practical deployment on edge devices.

---

## **Main Critiques and Our Responses**
1. **Function-Preservation Guarantees**
   *Critique:* Unclarity whether transforms remain function-preserving after training (h3nK, GL33, Hwow).
   *Response:* Yes, all are function-preserving, even after training. Highlighted formal proofs (Section 3.1), clarified transform $T_v$ guarantees and how parameterizations enforce function-preservation throughout training. Unit tests and new perturbation experiment (**Appendix L**) show function-preservation empirically.

2. **Residual Scaling Intuition and Stability**
   *Critique:* Lack of visualization and stability analysis of $S_n$ (9RjL, h3nK).
   *Response:* Added **Figure 2** illustrating outlier mitigation. New sensitivity analysis across model sizes confirms numerical stability (**Appendix L**).

3. **Local Optimization Justification**
   *Critique:* Why L₄ norm and its effect on convergence (h3nK, GL33).
   *Response:* Added ablation (**Appendix F.2.1**) comparing $L_p$ norms used in local optimization; all improve stability versus no optimization, with negligible overhead (<1% training time).

4. **Training Cost, Stability, and Scalability**
   *Critique:* Two-stage optimization may increase cost (h3nK, GL33), and concerns about scalability (HwoW)
   *Response:* Training time is comparable to SpinQuant and up to **45% faster than FlatQuant** (**Appendix J**). Training is more stable than baselines (**Appendix F.2 and L**). Memory overhead is zero since teacher and student share weights. Local optimization has negligible overhead (<1%). Added results for Qwen 2.5 **32B** to show scalability (**Table 2**).

5. **Comparison with OSTQuant**
   *Critique:* No OSTQuant results (Hwow, GL33).
   *Response:* Added OSTQuant to main results (**Tables 2,3**). FPTQuant consistently outperforms OSTQuant on static quantization and is comparable under dynamic settings while using fewer transforms. OSTQuant is between SpinQuant/FlatQuant in terms of cost/performance/release data, but not function-preserving (**Appendix L**).

6. **FlatQuant is Better in Some Quantization Settings**
   *Critique:* Accuracy gap with FlatQuant under W4A4KV4 (Hwow, GL33).
   *Response:* Clarified speed-accuracy trade-off, **FlatQuant is up to 29% slower**, and referred to **Appendix F, Table 12** that demonstrates adding one non-mergeable transform narrows the gap significantly. Overall, FPTQuant offers the best accuracy–efficiency trade-off.

7. **Extra models**
   *Critique:* More models (h3nK).
   *Response:* Added Ministral-8B and Qwen 2.5 32B models (**Table 2**).

---

## **Why This Paper Matters to ICLR**
FPTQuant formalizes the concept of function-preserving transforms and demonstrates how principled design can deliver **state-of-the-art efficiency/performance** with minimal engineering overhead. It achieves a Pareto-optimal balance of **accuracy, efficiency, and deployability**, addressing a core challenge in scaling LLM inference. These contributions align strongly with ICLR’s mission to advance both theoretical understanding and practical impact in machine learning.

Thank you for your consideration.

**Best regards,**
Authors of 16701

---

### Meta-Review · Area_Chair_Nv1j · 2025-12-16

**Summary:**

This paper proposes a framework for applying function-preserving transforms to LLM quantization, where three specific FPTs are proposed to better balance accuracy and compression. The AC individually appreciates this framework.

After reading the paper and rebuttal, the AC is inclined to reject this paper. The decision is based on the following judgments:
1. There are remaining unresolved concerns (risk of overfitting), and three active reviewers still express concerns (such as contribution and comparison with FLATQuant) about the submission.
2. Another major weakness is that the proposed method still cannot beat FLATQuant on many benchmarks. Although the former has some advantage in training cost, this advantage is not that critical for this topic.

The authors are encouraged to include these modifications for a future submission.

**Reviewer Concerns:**

1. Reviewer 9RjL (partly solved): The concern  "Risk of overfitting when using scale to train quantization coefficients statically" still exits.
2. Reviewer h3nK (partly solved): responded to the authors but proposed several further concerns, and the AC thinks the author still partly addresses these concerns.
3. Reviewer Hwow (partly solved):  still had unresolved concerns about Innovation and Theoretical Foundation and Engineering Implementation and Performance.
4. Reviewer GL33 (mostly solved): However, the reviewer still expressed concern about the current contributions.

**Reviewer Scores:**

1. Reviewer 9RjL would keep 4  (likely raised to 6 in best).
2. Reviewer h3nK decided to keep 4 (likely raised to 6 in best).
3. Reviewer Hwow decided to keep 4.
4. Reviewer GL33 decided to keep 4.

---

### Decision · Program_Chairs · 2026-01-26

Reject